# RoPE Attention Can Be Trained in Almost Linear Time

## Abstract

The Rotary Position Embedding (RoPE) mechanism has become a powerful enhancement to the Transformer architecture, which enables models to capture token relationships when encoding positional information. However, the RoPE mechanisms make the computations of attention mechanisms more complicated, which makes efficient algorithms challenging. Earlier research introduced almost linear time algorithms for the forward computation under specific parameter settings of bounded entries (i.e., in time $n^{1+o(1)}$ where $n$ is the number of input tokens), but has not addressed backward computation. In this work, we develop the first almost linear time algorithm for backward computations in the RoPE-based attention under bounded entries. Our approach builds on recent advancements in fast RoPE attention computations, utilizing a novel combination of the polynomial method and the Fast Fourier Transform. Furthermore, we show that with lower bounds derived from the Strong Exponential Time Hypothesis (SETH), the bounded entry condition is necessary for subquadratic performance.

## 1 Introduction

The GPT-o3 (OpenAI, 2024), Llama 3.3 (Llama Team, 2024; AI, 2024), Claude 3.5 (Anthropic, 2024b) are transformed-based Large Language Models (LLMs), have become important tools in natural language processing, which enables applications from machine translation to sentiment analysis. In the Transformer architecture, attention mechanisms, computationally intensive operations, compute token correlations within the sequence (Vaswani et al., 2017). The efficiency of attention computations, both in forward computations and backward gradient computations, directly influenced the scalability and feasibility of training LLMs, especially when the size and input context length of these LLMs continue to grow (Alman & Song, 2024a; 2023). In recent research, Rotating Position Embedding (RoPE) (Su et al., 2024) has become a popular modification to the attention mechanism, and it has enabled models to capture positional relationships between tokens with better expressiveness. The RoPE mechanism has been adopted in state-of-the-art models, such as Llama (Touvron et al., 2023a;b; Llama Team, 2024), Claude (Anthropic, 2024b), Apple's LLMs (Gunter et al., 2024; McKinzie et al., 2024), and many others, but the implementation of RoPE complicates attention computation due to the additional structure imposed by position-dependent rotations (Su et al., 2024). In recent work, (Alman & Song, 2024b) demonstrated an efficient algorithm for forward computation of RoPE attention in the bounded entry regime. However, backward computation, the process of calculating gradients for model optimization, has been explored less.

Backward computation introduces additional complexity because it requires the evaluation of gradients that involve non-linear transformations of the attention matrix and positional embeddings. In (Alman & Song, 2023), they present their algorithm to approximate forward computations of fast attention with bounded entries using the polynomial methods and low-rank approximation. In (Alman & Song, 2024c), they propose almost linear time, i.e., $n^{1+o(1)}$ where $n$ is the number of input tokens, an algorithm to compute backward gradients for fast attention with bounded entries. In recent work, (Alman & Song, 2024b) proposes efficient algorithm to perform the forward computation of RoPE-based attention using the polynomial methods and Fast Fourier Transform. Therefore, it is natural to raise the key question:

*Can backward computations for the RoPE attention match the efficiency of their forward computations in the bounded entry regime?*

In this work, we aim to address the question by presenting the first efficient algorithm for backward computation in RoPE attention under the bounded entry. Our main result shows that the backward gradient computations for the RoPE attention match their forward version's efficiency. Therefore, by leveraging our algorithm in approximating backward computations in the RoPE attention with the forward algorithm from (Alman & Song, 2024b), we will improve the overall time complexity of RoPE attention to almost linear time with bounded entries.

To the best of our knowledge, this is the first work to characterize the fine-grained complexity of backward computations in RoPE attentions, extending prior results on forward computations in RoPE attention (Alman & Song, 2024b). Our contribution can be described as follows.

- We formulated the closed-form gradient for the RoPE attention (see Lemma 4.1) along with its exact time complexity (see Theorem 4.2).
- We derive the almost linear time backward approximation (see Theorem 5.7) for RoPE attention based on the closed-form gradient.
- We show that with lower bounds derived from the SETH, the bounded entry condition is necessary for subquadratic performance (see Theorem 6.1).

**Roadmap.** In Section 2, we present some relevant papers. In Section 3, we show essential components for the RoPE attention. Section 4 gives the closed form of the RoPE Attention gradient and discusses the exact time complexity to compute it. Section 5 shows the fast computation of the RoPE Attention gradient in almost linear time. Section 6 details the lower bounds of hardness. Finally, Section 7 provides conclusions and avenues for future work.

## 2 RELATED WORK

**Rotary Position Embedding.** At a high level, RoPE gives more expressive power to the model in exchange for making the computational problem more complicated. In particular, many prior algorithms, such as the algorithm of (Alman & Song, 2023), no longer apply to RoPE for fundamental reasons we will discuss. RoPE was proposed by (Su et al., 2024) and has been used extensively in large-scale industrial models. Examples which are known to use RoPE include the open-source models released by Meta such as Llama (Touvron et al., 2023a) (see page 3), Llama 2 (Touvron et al., 2023b) (see page 5), Llama 3 (Llama Team, 2024) (see page 7), and the close-source LLM Claude 3.5 (Anthropic, 2024b) released by Anthropic. Apple also incorporates RoPE into their LLM architecture (see (McKinzie et al., 2024), and page 3 of (Gunter et al., 2024)).

**Fast Attention Computation.** The attention mechanism has often been criticized for its quadratic computational complexity concerning context length, a challenge that becomes more pronounced as the sequence length grows in today's LLMs (Achiam et al., 2023; OpenAI, 2024; Llama Team, 2024; AI, 2024; Anthropic, 2024a;b). However, this issue can be addressed using polynomial kernel approximation methods (Aggarwal & Alman, 2022), which facilitate constructing the approximated attention matrix using low-rank approximations. Such techniques enable substantial improvements in computation speed, allowing a single attention layer to perform both training and inference nearly as fast as linear time (Alman & Song, 2023; 2024c). (Liang et al., 2024b) further extends this efficiency to support multi-layer transformer architectures for both training and inference. In addition, these techniques can generalize to advanced attention mechanisms, such as tensor attention, while preserving the almost linear time complexity in both training and evaluation phases (Alman & Song, 2024a; Liang et al., 2024c). Beyond this, alternative theoretical methods also exist. For example, the conv-basis approach introduced in (Liang et al., 2024a) offers another avenue for speeding up attention computation.

**Gradient Approximation.** Using low-rank approximation to approximate the gradient is a common approach for optimizing the training of transformers by reducing the complexity in the computations, such as (Liang et al., 2024b;c; Alman & Song, 2024c; Hu et al., 2024). Specifically, (Alman & Song, 2024c) extends the low-rank approximation technique developed in (Alman & Song, 2023), which studies the forward computation of attention to approximate the gradient of the attention computation. In (Liang et al., 2024b), they further develop the low-rank approximation technique in (Alman & Song, 2024c) to study multi-layer transformers, showing they can use nearly

linear time to approximate the backward computations of multi-layer transformers. On the other hand, (Liang et al., 2024c) generalizes the gradient approximation of (Alman & Song, 2024c) to another direction: they use it to study the training of the tensor version of attention computation that develops from the forward computation as in (Alman & Song, 2024a). Finally, (Hu et al., 2024) leverages the low-rank approximation technique to study the training of Diffusion Transformers (DiTs).

## 3 PRELIMINARIES ON ROPE ATTENTION

In Section 3.1, we talk about the notation and foundational concepts. In Section 3.2, we formalize our problems. In Section 3.3, we talk about the reformulation of the loss function using the tensor trick and analyze the computational complexity of the reformulated expressions.

### 3.1 NOTATION

For $n \in \mathbb{Z}^+ \cup \{0\}$, for set $\{1, 2, \cdots, n\}$, we denote the set by using the notation $[n]$. Here, we define the concept of nearly linear time when the time is $O(n \log n)$. We introduce the concept of almost linear time when time is $O(n^{1+o(1)})$. Given $a$ as any vector, we say the diagonal matrix of $c$ is $\mathrm{diag}(c)$ where $c_i$ means the $i, i$-th entry in the matrix $\mathrm{diag}(c)$. For any matrix, we denote the support of the matrix using the notation supp, that is, the set of entries where the matrix is nonzero. $B^\top$ is defined as $(B^\top)_{i,j} := B_{j,i}$. Suppose there are two vectors $c, d$ of the same length. We denote the entry-wise multiplication using the notation $c \circ d$; that is, the $i$-th entry in that vector is $c_i d_i$. To denote the Frobenius norm, for any matrix $B$, we denote it as $\|B\|_F := \sqrt{\sum_{i,j} B_{i,j}^2}$; to denote the maximum norm of matrix $B$, we use $\|B\|_\infty := \max_{i,j} |B_{i,j}|$. Suppose there are two matrices $C, D$ of the same dimensions. We represent the Hadamard product or the entry-wise multiplication by using the notation $C \circ D$, that is, $(i, j)$-th entry of the matrix is $C_{i,j} \cdot D_{i,j}$. Let $C \in \mathbb{R}^{n_0 \times m_0}$ and $D \in \mathbb{R}^{n_1 \times m_1}$. We define $C \otimes D$ is an $n_0 n_1 \times m_0 m_1$ matrix, where $(C \otimes D)_{(j_0-1)n_1+j_1,(i_0-1)m_2+i_1}$ is equal to $C_{j_0,i_0} D_{j_1,i_1}$ for any $j_0 \in [n_0], i_0 \in [m_0], j_1 \in [n_1], i_1 \in [m_1]$.

### 3.2 PROBLEM DEFINITION

Let $n$ be the number of input tokens, and let $d$ be the hidden/feature dimensions. We state the generalization of the standard RoPE attention from (Alman & Song, 2024b).

**Definition 3.1** (A General Approximate RoPE Attention Computation, ARAttC, Definition 1.1 in (Alman & Song, 2024b))**.** *Let $B > 0$ and $\epsilon > 0$ denote two parameters. Given a set of matrices $W_{-(n-1)}, \cdots, W_{-1}, W_0, W_1, \cdots, W_{n-1} \in \mathbb{R}^{d \times d}$ where $\mathrm{supp}(W_i) \subset S$ for all $i \in \{-(n-1), \cdots, -1, 0, 1, \cdots, n-1\}$. Here $S \subseteq [d] \times [d]$ where $|S| = O(d)$. Given three $n \times d$ matrices $Q, K, V$ with the guarantee that $\|Q\|_\infty, \|K\|_\infty, \|V\|_\infty \leq B$ and $\|W\|_\infty \leq 1$. We define matrix $A \in \mathbb{R}^{n \times n}$ as, for $i, j \in [n]$, $A_{i,j} := \exp(Q_{i,*} W_{i-j} K_{j,*}^\top / d)$. We define $D := \mathrm{diag}(A\mathbf{1}_n)$. The goal of General Approximate RoPE Attention Computation is to output a matrix $T \in \mathbb{R}^{n \times d}$ such that $\|T - \mathsf{ARAttC}\|_\infty \leq \epsilon$ is small, where $\mathsf{ARAttC} := D^{-1}AV$. For matrix $M$, we use $\|M\|_\infty := \max_{i,j} |M_{i,j}|$. Note that the $1/d$ factor inside $\exp$ in the definition of $A$ is a normalization factor.*

Our focus is to find weights to fit the attention to a desired output. Let $Q := A_1 X_1$, $K := A_2 X_2$, and $V := A_3 Y$. We use $X_1$, $X_2$, and $X_3$ to represent the weights $W_Q$, $W_K$ and $W_V$, respectively. We use $A_1$, $A_2$, and $A_3$ to replace the input matrix to handle the more general settings such as cross attention. Then, the attention matrix is as follows.

$$A(X_1, X_2)_{i,j} := \exp((A_1 X_1)_{i,*} W_{i-j} (A_2 X_2)_{j,*}^\top / d)$$
$$= \exp(A_{1,i,*} X_1 W_{i-j} X_2^\top A_{2,j,*}^\top / d).$$

We define $w_{i-j} := \mathrm{vec}(W_{i-j}) \in \mathbb{R}^{d^2}$ and define $\mathsf{W}$ such that $\mathsf{W}_{j_0,*}$ is an $1 \times d^2$ block and $\mathsf{W}_{i+(j-1)n,*} := w_{i-j}^\top$. Here, let $\mathsf{A} := A_1 \otimes A_2 \in \mathbb{R}^{n^2 \times d^2}$ and $\mathsf{X} := X_1 \otimes X_2 \in \mathbb{R}^{d^2 \times d^2}$. We can show that

$$A_{1,i,*} X_1 W_{i-j} X_2^\top A_{2,j,*}^\top$$

$$= (A_{1,i,*} \otimes A_{2,j,*})(X_1 \otimes X_2) \operatorname{vec}(W_{i-j})$$
$$= \mathsf{A}_{i+(j-1)n,*} \mathsf{X} w_{i-j},$$

where the first step uses the tensor trick, and the second step uses the definitions of $w_{i-j}, \mathsf{A}$, and $\mathsf{X}$. Thus we can reformulate the attention matrix $A$ as, for $i, j \in [n]$

$$A(\mathsf{X})_{i,j} = \exp(\underbrace{\mathsf{A}_{i+(j-1)n,*}}_{1 \times d^2} \underbrace{\mathsf{X}}_{d^2 \times d^2} \underbrace{w_{i-j}/d}_{d^2 \times 1}).$$

Using the tensor trick again, we have

$$A(\mathsf{X})_{i,j} = \exp(\underbrace{(\mathsf{A}_{i+(j-1)n,*} \otimes w_{i-j}^\top)}_{1 \times d^4} \underbrace{\operatorname{vec}(\mathsf{X})/d}_{d^4 \times 1})$$
$$= \exp(\underbrace{(\mathsf{A}_{i+(j-1)n,*} \otimes \mathsf{W}_{i+(j-1)n,*})}_{1 \times d^4} \underbrace{\operatorname{vec}(\mathsf{X})/d}_{d^4 \times 1}).$$

Hence, by definition of row-wise Kronecker product, we have

$$\operatorname{vec}(A(\mathsf{X})) = \exp(\underbrace{(\mathsf{A} \oslash \mathsf{W})}_{n^2 \times d^4} \underbrace{\operatorname{vec}(\mathsf{X})/d}_{d^4 \times 1}).$$

We define the matrix $D(\mathsf{X}) \in \mathbb{R}^{n \times n}$ as

$$D(\mathsf{X}) = \operatorname{diag}(\underbrace{A(\mathsf{X})}_{n \times n} \underbrace{\mathbf{1}_n}_{n \times 1}).$$

Then, the optimization problem in the context of RoPE attention computation is described as follows:

**Definition 3.2** (Optimize RoPE Attention). *Let $B > 0$ and $\epsilon > 0$ denote two parameters. Given a set of matrices $W_{-(n-1)}, \cdots, W_{-1}, W_0, W_1, \cdots, W_{n-1} \in \mathbb{R}^{d \times d}$ where $\operatorname{supp}(W_i) \subset S$ for all $i \in \{-(n-1), \cdots, -1, 0, 1, \cdots, n-1\}$. Here $S \subseteq [d] \times [d]$ where $|S| = O(d)$. For $i, j \in [n]$, let $\mathsf{W} \in \mathbb{R}^{n^2 \times d^2}$ such that $\mathsf{W}_{i+(j-1)n,*} = \operatorname{vec}(W_{i-j})$. Here, we suppose four $n \times d$ matrices $A_1, A_2, A_3, E$, and we have three $d \times d$ matrices $X_1, X_2, Y$. Let $\mathsf{X} := X_1 \otimes X_2 \in \mathbb{R}^{d^2 \times d^2}$. We define the matrix $A(\mathsf{X}) \in \mathbb{R}^{n \times n}$ as the matrix representation of $\exp(\underbrace{(\mathsf{A} \oslash \mathsf{W})}_{n^2 \times d^4} \underbrace{\operatorname{vec}(\mathsf{X})/d}_{d^4 \times 1})$ and the $n \times n$ matrix $D(\mathsf{X}) := \operatorname{diag}(\underbrace{A(\mathsf{X})}_{n \times n} \underbrace{\mathbf{1}_n}_{n \times 1})$. The RoPE attention optimization problem $\min_{\mathsf{X} \in \mathbb{R}^{d^2 \times d^2}} \mathsf{Loss}(\mathsf{X})$ is formulated as follows:*

$$\min_{\mathsf{X} \in \mathbb{R}^{d^2 \times d^2}} 0.5 \|D(\mathsf{X})^{-1} A(\mathsf{X}) A_3 Y - E\|_F^2.$$

Note that we are able to get the gradient computation of $\mathsf{Loss}$ with respect to $X_1$ or $X_2$ based on the chain rule because

$$\frac{\mathrm{d}\mathsf{Loss}(X_1, X_2)}{\mathrm{d}X_1} = \frac{\mathrm{d}\mathsf{Loss}(\mathsf{X})}{\mathrm{d}\mathsf{X}} \frac{\mathrm{d}\mathsf{X}}{\mathrm{d}X_1}$$
$$= \frac{\mathrm{d}\mathsf{Loss}(\mathsf{X})}{\mathrm{d}\mathsf{X}} \frac{\mathrm{d}(X_1 \otimes X_2)}{\mathrm{d}X_1}$$
$$= \frac{\mathrm{d}\mathsf{Loss}(\mathsf{X})}{\mathrm{d}\mathsf{X}} (I_{d \times d} \otimes X_2).$$

Our approximation task can be formalized as follows.

**Definition 3.3** (The Approx of the gradient of RoPE Attention Loss Function, $\mathsf{ARAttLGC}(n, d, B, \epsilon)$). *Let $B > 0$ and $\epsilon > 0$ denote two parameters. Given a set of matrices $W_{-(n-1)}, \cdots, W_{-1}, W_0, W_1, \cdots, W_{n-1} \in \mathbb{R}^{d \times d}$ where $\operatorname{supp}(W_i) \subset S$ for all $i \in \{-(n-1), \cdots, -1, 0, 1, \cdots, n-1\}$. Here $S \subseteq [d] \times [d]$ where $|S| = O(d)$. For $i, j \in [n]$, let $\mathsf{W} \in \mathbb{R}^{n^2 \times d^2}$ such that $\mathsf{W}_{i+(j-1)n,*} = \operatorname{vec}(W_{i-j})$. Let $X_1, X_2, Y \in \mathbb{R}^{d \times d}$.*

*Let $\mathsf{X} := X_1 \otimes X_2 \in \mathbb{R}^{d^2 \times d^2}$. We have four $n \times d$ matrices Let $A_1, A_2, A_3, E$. Let $\mathsf{A} \in \mathbb{R}^{n^2 \times d^2}$ such that $\mathsf{A}$ equals to an $n^2 \times d^2$ matrix from $A_1 \otimes A_2$. Assume $\|A_1 X\|_\infty \leq B, \|A_2 X\|_\infty \leq B, \|A_3 Y\|_\infty \leq B, \|W\|_\infty \leq 1$. Assume that all the $\log(n)$ bits model is applied throughout all numbers in matrices. We define $\mathsf{Loss}(\mathsf{X})$ from Def. 3.2. Here, we define $\frac{d\mathsf{Loss}(\mathsf{X})}{d\mathsf{X}}$ as the loss function gradient. Then, our target is to output a vector $\widetilde{g} \in \mathbb{R}^{d^4}$ satisfying:*

$$\|\widetilde{g} - \frac{d\mathsf{Loss}(\mathsf{X})}{d\mathsf{X}}\|_\infty \leq \epsilon.$$

### 3.3 Reformulation of the Loss Function

In this section, we are able to reformulate and simplify the loss function based on the definitions provided in Section B. This reformulation provides a structured representation of the loss in terms of its components, using the tensor trick to simplify computations and facilitate analysis.

The following lemma formalizes this reformulation, consolidating the expressions for the loss function and connecting its components:

**Lemma 3.4** (Loss Function Formulation). *Given three $n \times d$ input sequence matrices $A_1$, $A_2$, and $A_3$, we define $\mathsf{A} = A_1 \otimes A_2 \in \mathbb{R}^{n^2 \times d^2}$ and $\mathsf{X} = X_1 \otimes X_2 \in \mathbb{R}^{d^2 \times d^2}$, where $\otimes$ denotes Kronecker product. Given $W$ is a $n^2 \times d^2$ matrix, we define $\widetilde{A} = A \oslash W$, where $\oslash$ is the row-wise Kronecker product from Fact A.5. Let $j_0 \in [n]$, we define $\widetilde{A}_{j_0} \in \mathbb{R}^{n \times d^2}$ be a block of size $n \times d^2$ from $\widetilde{A}$. Let $E \in \mathbb{R}^{n \times d}$ be a matrix, for $j_0 \in [n]$ and $i_0 \in [d]$, we define $E_{j_0, i_0}$ as the $(j_0, i_0)$-th entry of the matrix $E$. We use $\mathsf{Loss}$ function from Definition 3.2. Based on Def. B.6, for $j_0$ in the set $[n]$ and $i_0$ in the set $[d]$, we get $\mathsf{Loss}(\mathsf{X})_{j_0, i_0}$. Then, we have*

$$\mathsf{Loss}(\mathsf{X}) = \sum_{j_0 \in [n]} \sum_{i_0 \in [d]} \mathsf{Loss}(x)_{j_0, i_0}.$$

*Proof.* We present the reformulation of the Loss Function using the tensor trick as follows.

$$\mathsf{Loss}(\mathsf{X}) = 0.5 \|D(\mathsf{X})^{-1} A(\mathsf{X}) A_3 Y - E\|_F^2$$

$$= \sum_{j_0=1}^{n} \sum_{i_0=1}^{d} 0.5 \cdot (\langle\langle \exp(\widetilde{\mathsf{A}}_{j_0} x), \mathbf{1}_n \rangle^{-1}$$

$$\exp(\widetilde{\mathsf{A}}_{j_0} x), A_3 Y_{*, i_0} \rangle - E_{j_0, i_0})^2$$

$$= \sum_{j_0=1}^{n} \sum_{i_0=1}^{d} 0.5 (\langle s(x)_{j_0}, v(y)_{i_0} \rangle - E_{j_0, i_0})^2$$

$$= \sum_{j_0=1}^{n} \sum_{i_0=1}^{d} \mathsf{Loss}(x)_{j_0, i_0}$$

where the 1st equality is based on Def. 3.2, the definition of Frobenius norm derives the 2nd equality, the 3rd equality is due to Def. B.3 and Def. B.4, and the 4th step is based on Def. B.6. □

## 4 Exact Gradient Computation Time

In this section, we provide the gradient computations of RoPE attentions. In Section 4.1, we formulate the gradient in its closed form. In Section 4.2, we conduct a time complexity analysis on the exact computation of RoPE attention gradients.

### 4.1 Reformulate the Gradient into Its Closed Form

In this section, we present the closed-form gradient of RoPE attention.

**Lemma 4.1** (Gradient Reformulation, $\frac{d\mathsf{Loss}(x)}{dx}$, Informal Version of Lemma C.4). *For every $i \in [d^4]$, we choose the following functions*

- *The Normalized Softmax function $s(x)_{j_0} \in \mathbb{R}^n$ (see Definition B.3),*

- *The Error term $\ell(x)_{j_0,i_0} \in \mathbb{R}$ (see Definition B.5),*

- *The Loss term $\mathsf{Loss}(x)_{j_0,i_0} \in \mathbb{R}$ (see Definition B.6),*

- *We define $\beta(x)_{j_0} \in \mathbb{R}^n$ is $\underbrace{A_3 Y}_{n \times d} \underbrace{\ell(x)_{j_0,*}^\top}_{d \times 1}$*

- *We define $\gamma(x)_{j_0} \in \mathbb{R}^n$ is $(\mathrm{diag}(s(x)_{j_0}) - s(x)_{j_0} s(x)_{j_0}^\top)\beta(x)_{j_0}$*

*Then, we get $\frac{\mathrm{dLoss}(x)}{\mathrm{d}x} = \underbrace{\widetilde{\mathsf{A}}^\top}_{d^4 \times n^2} \mathrm{vec}(\underbrace{\gamma(x)}_{n \times n})$.*

*Proof.* See full proof at Lemma C.4. $\qquad\qquad\square$

## 4.2 Time Complexity for Computing the Gradient of RoPE Attention

In this section, we provide the time complexity of computing the exact gradient of RoPE attention.

**Theorem 4.2** (RoPE attention gradient computation time complexity)**.** *We define three $n \times d$ input matrices as $A_1, A_2, A_3$, and the $n \times d$ approximated attention computation matrix as $E$. We define several input fixed matrices as $X_1, X_2, Y \in \mathbb{R}^{d \times d}$. We define $\mathsf{X} = X_1 \otimes X_2$, $\mathsf{A} = A_1 \otimes A_2$. We define $x := \mathrm{vec}(\mathsf{X})$ and try to get the Loss function gradient. Let $g := \frac{\mathrm{dLoss}(X_1,X_2)}{\mathrm{d}x}$ where $\mathsf{Loss}(X_1, X_2)$ from Def. 3.3. Then, it costs $O(\mathcal{T}_{\mathrm{mat}}(n, d, d) + \mathcal{T}_{\mathrm{mat}}(n, d, n))$ time to get the gradient $g \in \mathbb{R}^{d^4}$.*

*Proof.* See full proof at Theorem C.8. $\qquad\qquad\square$

Note that $O(\mathcal{T}_{\mathrm{mat}}(n, d, d) + \mathcal{T}_{\mathrm{mat}}(n, d, n)) \geq \Omega(n^2)$. Thus, the naive RoPE attention gradient computation is a complexity obstacle in practice, as discussed in Section 1. Based on the closed formulation in Lemma 4.1, we derive our acceleration method, which will be introduced in the following section.

## 5 Compute RoPE Attention Gradient in Almost Linear Time

In this section, we present our main result. With the low-rank approximation, we can approximate the RoPE gradient computations in almost linear time.

In Section 5.1, we discuss the techniques we used to develop the almost linear time algorithm. In Section 5.2, we provide the proof of approximating $s(x)$ in almost linear time. In Section 5.3, we give the proof to approximate the error term $\ell(x)$. In Section 5.4, we show how to approximate $\beta(x)$ in almost linear time. In Section 5.5, we present our technique to approximate $\gamma(x)$. In Section 5.6, we present our main results, which compute RoPE Attention gradient in almost linear time.

### 5.1 Technique Overview

In recent work (Alman & Song, 2024b), they present an almost linear-time algorithm to compute forward computations of RoPE attention as follows.

**Lemma 5.1** (Theorem 1.3 in (Alman & Song, 2024b))**.** *Suppose $d = O(\log n)$ and $B = o(\sqrt{\log n})$. There is an $n^{1+o(1)}$ time algorithm to approximate $\mathsf{ArAttC}$ up to $\epsilon = 1/\mathrm{poly}(n)$ additive error.*

Recall that the closed form gradient of RoPE attention is $\frac{\mathrm{dLoss}(x)}{\mathrm{d}x} = \widetilde{\mathsf{A}}^\top \mathrm{vec}(\gamma(x))$ from Lemma 4.1. We need to show $\gamma(x)$ can be low-rank approximated in $O(n^{1+o(1)})$ time with $1/\mathrm{poly}(n)$ error.

To low rank approximate $\gamma(x)$, we use the strategy to split $\gamma(x)$ into two terms, $\gamma_1(x)$ and $\gamma_2(x)$, and run the approximation separately. From Lemma 4.1, $\gamma(x)_{j_0} \in \mathbb{R}^n$ is $(\mathrm{diag}(s(x)_{j_0}) -$

$s(x)_{j_0} s(x)_{j_0}^\top) \beta(x)_{j_0}$. We define $\gamma_1(x)_{j_0} = \mathrm{diag}(s(x)_{j_0}) \beta(x)_{j_0}$ and $\gamma_2(x)_{j_0} = s(x)_{j_0} s(x)_{j_0}^\top \beta(x)_{j_0}$; thus, we can have $\gamma(x) = \gamma_1(x) - \gamma_2(x)$.

In the definitions of $\gamma_1(x)$ and $\gamma_2(x)$ provided above, they both contain $s(x)$ and $\beta(x)$. In order to find the almost linear time algorithm of $\gamma(x)$, we need to first show that there exists $O(n^{1+o(1)})$ time complexity approximation for $s(x)$ and $\beta(x)$ with $\epsilon/\mathrm{poly}(n)$ error first. From Lemma 4.1, we have $\beta(x)_{j_0} \in \mathbb{R}^n$ is $A_3 Y \ell(x)_{j_0,*}^\top$. Based on the $\beta(x)$ definition, we need to show $\ell(x)$ can be approximated in almost linear time first.

Overall, to develop the $O(n^{1+o(1)})$ time complexity algorithm to compute RoPE gradients with $\epsilon/\mathrm{poly}(n)$ error, we need to prove the existence of almost linear time algorithms for $s(x), \ell(x), \gamma(x)$, and $\beta(x)$ with low rank approximation.

## 5.2 Approximate $s$ Using Low Rank Approximation

In this section, we use the low-rank approximation technique to approximate the normalized Softmax $s(x)$ in almost linear time.

**Lemma 5.2** (Low Rank Approximate $s(x)$). *For any $B = o(\sqrt{\log n})$, let $k_1$ equals to $n^{o(1)}$ such that: Suppose we have two $n \times d$ matrices $A_1, A_2$, $X_1, X_2 \in \mathbb{R}^{d \times d}$ and $\mathsf{X} = X_1 \otimes X_2 \in \mathbb{R}^{d^2 \times d^2}$. Assume we can use $O(\log n)$ bits to write every entry from $s(x)$. It holds that $\max\{\|A_1 X_1\|_\infty, \|A_2 X_2\|_\infty\} \le B$, then there are three matrices $U_1, V_1, W_1 \in \mathbb{R}^{n \times k_1}$ such that $\|U_1 V_1^\top - s(x)\|_\infty \le \epsilon/\mathrm{poly}(n)$. Here $s(x) = D^{-1} A \in \mathbb{R}^{n \times n}$ where $A$ is defined as the matrix representation of $\exp((\mathsf{A} \oslash \mathsf{W}) \mathrm{vec}(X))$, and $D = \mathrm{diag}(A/d)\mathbf{1}_n$. Moreover, these matrices $U_1, V_1$ can be created explicitly in $n^{1+o(1)}$ time.*

*Proof.* By definition of $A(\mathsf{X})$, we have $\mathrm{vec}(A(\mathsf{X})) = \exp(\mathsf{A} \oslash \mathsf{W}) \mathrm{vec}(X)$.

Hence, using the tensor trick, we have

$$A(\mathsf{X})_{i,j} = \exp((\mathsf{A}_{i+(j-1)n} \otimes \mathsf{W}_{i+(j-1)n}) \mathrm{vec}(\mathsf{X})/d)$$
$$= \exp((\mathsf{A}_{i+(j-1)n} \otimes w_{i-j}^\top) \mathrm{vec}(\mathsf{X})/d).$$

We define $w_{i-j} := \mathrm{vec}(W_{i-j}) \in \mathbb{R}^{d^2}$ and define $\mathsf{W}$ such that $\mathsf{W}_{j_0}$ is an $1 \times d^2$ block and $\mathsf{W}_{i+(j-1)n} := w_{i-j}^\top$. We also define $\mathsf{A} := A_1 \otimes A_2 \in \mathbb{R}^{n^2 \times d^2}$ and $\mathsf{X} := X_1 \otimes X_2 \in \mathbb{R}^{d^2 \times d^2}$. We use $\mathsf{A}_{j_0}$ to denote the a $1 \times d^2$ subblock of $\mathsf{A}$.

We can reformulate the attention matrix $A$ as, for $i, j \in [n]$

$$A(\mathsf{X})_{i,j} = \exp(\underbrace{\mathsf{A}_{i+(j-1)n}}_{1 \times d^2} \underbrace{\mathsf{X}}_{d^2 \times d^2} \underbrace{w_{i-j}/d}_{d^2 \times 1}).$$

Thus, we can show that $\mathsf{A}_{i+(j-1)n} \mathsf{X} w_{i-j} = (A_{1,i,*} \otimes A_{2,j,*})(X_1 \otimes X_2) \mathrm{vec}(W_{i-j}) = A_{1,i,*} X_1 W_{i-j} X_2^\top A_{2,j,*}^\top$, where 1st equality uses definitions of $w_{i-j}, \mathsf{A}$, and $\mathsf{X}$, and the second step uses the tensor trick. We complete our proof after applying Lemma 5.1. $\square$

## 5.3 Approximate $\ell$ Using Low Rank Approximation

In this section, we use the low-rank approximation technique to approximate $\ell(x)$

**Lemma 5.3** (Low Rank Approximate $\ell(x)$). *Let $d$ equal $O(\log n)$. Suppose we can use $O(\log n)$ bits to write every entry in $E, v(y) \in \mathbb{R}^{n \times d}$. Define the $\ell(x) \in \mathbb{R}^{n \times d}$ as specified in Def. B.5. Then, we have $U_1, V_1 \in \mathbb{R}^{n \times k_1}$ such that $\|U_1 V_1^\top v(y) - E - \ell(x)\|_\infty \le \epsilon/\mathrm{poly}(n)$.*

*Proof.* Here, we present the bound as follows.

$$\|U_1 V_1^\top v(y) - E - \ell(x)\|_\infty = \|U_1 V_1^\top v(y) - s(x) v(y)\|_\infty$$
$$= \|v(y)\|_\infty \cdot \|U_1 V_1^\top - s(x)\|_\infty$$
$$\le \epsilon/\mathrm{poly}(n),$$

where the 1st is because of Def. B.5, 2nd step is based on the distributive law, and 3rd step is due to Lemma 5.2. $\square$

## 5.4 APPROXIMATE $\beta$ USING LOW RANK APPROXIMATION

In this section, we use the low-rank approximation technique to approximate $\beta(x)$

**Lemma 5.4** (Low Rank Approximate $\beta(x)$). *Let $k_2 = n^{o(1)}$. We define $\ell(x) \in \mathbb{R}^{n \times d}$ based on Def. B.5, and $v(y) \in \mathbb{R}^{n \times d}$ based on Def. B.4. We suppose $\beta(x)$ is equal to $v(y)\ell(x)^\top$, which is an $n \times n$ matrix. Let $U_2, V_2 \in \mathbb{R}^{n \times k_2}$ such that $\|U_2 V_2^\top - \beta(x)\|_\infty \leq \epsilon / \operatorname{poly}(n)$. In $n^{1+o(1)}$ time, we can get $U_2, V_2$.*

*Proof Sketch.* We define $\widetilde{\beta}(x) \approx \beta(x)$ and $\widetilde{\beta}(x) = v(y)v(y)^\top V_1 U_1^\top - v(y)E^\top$. We can first compute $v(y)^\top V_1$ as it can be computed in $n^{1+o(1)}$ time. Given all low rank matrices, we can have $U_2, V_2$ where $k_2 = \max\{d, k\} + d = n^{o(1)}$. Then we can compute $\|\widetilde{\beta}(x) - \beta(x)\|_\infty \leq \epsilon / \operatorname{poly}(n)$. (See full proof at Lemma D.3) $\qquad\square$

## 5.5 APPROXIMATE $\gamma$ USING LOW RANK APPROXIMATION

In this section, we use the low-rank approximation technique to approximate $\gamma(x)$. Specifically, we apply the polynomial methods to $\gamma_1(x)$ and $\gamma_2(x)$ where $\gamma(x) = \gamma_1(x) - \gamma_2(x)$.

First, we show the low-rank approximation of $\gamma_1(x)$.

**Lemma 5.5** (Low Rank Approximate $\gamma_1(x)$). *Let $k_1 = n^{o(1)}$. Let $k_2 = n^{o(1)}$. We suppose $\gamma_1(x)$ is $\operatorname{diag}(s(x))\beta(x)$, and $U_1, V_1$ be two $n \times k_1$ matrices, in which $\|U_1 V_1^\top - f(x)\|_\infty \leq \frac{\epsilon}{\operatorname{poly}(n)}$. We suppose two $n \times k_2$ matrices $U_2, V_2$ in which $\|U_2 V_2^\top - \beta(x)\|_\infty \leq \frac{\epsilon}{\operatorname{poly}(n)}$. Then we have two $n \times k_3$ matrices in which $\|U_3 V_3^\top - \gamma_1(x)\|_\infty \leq \epsilon / \operatorname{poly}(n)$. We can construct $U_3, V_3$ in $n^{1+o(1)}$ time.*

*Proof Sketch.* Let $U_3 = U_1 \oslash U_2$ and $V_3 = V_1 \oslash V_2$, and we can use $n^{1+o(1)}$ time to get them. From Lemma 5.2 and Lemma 5.4, we have $\widetilde{s}(x) = U_1 V_1^\top$ and $\widetilde{\beta}(x) = U_2 V_2^\top$. Based on Fact A.5, we can compute $\|U_3 V_3^\top - \gamma_1(x)\|_\infty \leq \|U_3 V_3^\top - \operatorname{diag}(s(x))\beta(x)\|_\infty \leq \frac{\epsilon}{\operatorname{poly}(n)}$ (See full proof at Lemma D.4). $\qquad\square$

Next, we show the low-rank approximation of $\gamma_2(x)$.

**Lemma 5.6** (Low Rank Approximate $\gamma_2(x)$). *Let $k_1 = n^{o(1)}$. Let $k_2 = n^{o(1)}$. Let $k_4 = n^{o(1)}$. Let $\gamma_2(x) \in \mathbb{R}^{n \times n}$ where for $j_0$ in set $[n]$, $j_0$ represents $j_0$-th column, $\gamma_2(x)_{j_0} = s(x)_{j_0} s(x)_{j_0}^\top \beta(x)_{j_0}$. We suppose $U_1, V_1 \in \mathbb{R}^{n \times k_1}$ in which $\|U_1 V_1^\top - s(x)\|_\infty \leq \frac{\epsilon}{\operatorname{poly}(n)}$. We suppose two $n \times k_2$ matrices $U_2, V_2$ in which $\|U_2 V_2^\top - \beta(x)\|_\infty \leq \frac{\epsilon}{\operatorname{poly}(n)}$. Then, we have $U_4, V_4 \in \mathbb{R}^{n \times k_4}$ such that $\|U_4 V_4^\top - \gamma_2(x)\|_\infty \leq \epsilon / \operatorname{poly}(n)$. We can get $U_4, V_4$ in $n^{1+o(1)}$ time.*

*Proof Sketch.* Let $\rho(x) \in \mathbb{R}^n$ be defined by $\rho(x)_{j_0} = s(x)_{j_0}\beta(x)_{j_0}$. We construct $\widetilde{\rho}(x)$ so that $(U_1 V_1)_{j_0,*}^\top \approx s(x)_{j_0}$ and $(U_2 V_2)_{j_0,*}^\top \approx \beta(x)_{j_0}$, implying $\widetilde{\rho}(x)_{j_0} = (U_1 V_1)_{j_0,*} \cdot (U_2 V_2)_{j_0,*}^\top$. Precomputing $V_1 V_2^\top$ takes $n^{1+o(1)}$ time, and then computing each $\widetilde{\rho}(x)_{j_0}$ costs $O(k_1 k_2)$, giving a total of $O(nk_1 k_2) = n^{1+o(1)}$. Next, we approximate $s(x)$ by $\widetilde{s}(x) = U_1 V_1^\top$ and define $\widetilde{\gamma}_2(x) = \widetilde{s}(x)\operatorname{diag}(\widetilde{\rho}(x))$; with $U_4 = U_1$ and $V_4 = \operatorname{diag}(\widetilde{\rho}(x))V_1$, we have $\widetilde{\gamma}_2(x) = U_4 V_4^\top$. To bound the error, note that $\|\widetilde{\gamma}_2(x) - \gamma_2(x)\|_\infty = \max_{j_0} \|\widetilde{s}(x)_{j_0} \widetilde{\rho}(x)_{j_0} - s(x)_{j_0} \rho(x)_{j_0}\|_\infty$ can be split and bounded via the triangle inequality so that $\|\widetilde{s}(x)_{j_0} - s(x)_{j_0}\|_\infty$ and $\|\widetilde{\rho}(x)_{j_0} - \rho(x)_{j_0}\|_\infty$ are small, leading to an overall error of at most $\epsilon / \operatorname{poly}(n)$, which completes the proof. (See full proof at Lemma D.5) $\qquad\square$

## 5.6 FAST COMPUTATION IN ALMOST LINEAR TIME

Based on Section 5.1, we have proved the almost linear time approximation of $s(x), \ell(x), \gamma(x)$, and $\beta(x)$ in Lemma D.1, D.2, D.3, D.5, and D.4. We are now ready to show our main result, which is to approximate RoPE gradient computation in almost linear time.

**Theorem 5.7** (Main result, Low Rank Approximate RoPE Attention Gradient). *Assuming the entries of $A_1, A_2, X_1, X_2, Y, E$ are represented using $O(\log n)$ bits, there is an $n^{1+o(1)}$ time algorithm to solve* $\mathsf{AAttLGC}(n, d = O(\log n), B = o(\sqrt{\log n}))$, *from Def. 3.3, with the accuracy upper bounded by $\frac{1}{\mathrm{poly}(n)}$ . To be more specific, a gradient vector $\widetilde{g} \in \mathbb{R}^{d^4}$ comes out of our algorithm where $\|\frac{\mathrm{dLoss}}{\mathrm{d}x} - \widetilde{g}\|_\infty \leq \frac{1}{\mathrm{poly}(n)}$.*

*Proof Sketch.* By Lemma D.5 and Lemma D.4, there exist matrices $\gamma_1(x)$ and $\gamma_2(x)$ such that $\gamma(x) = \gamma_1(x) - \gamma_2(x)$. We assume these lemmas follow from the low-rank approximations in Lemmas D.1–D.3, allowing us to write $\widetilde{\gamma}_1(x) = U_3 V_3^\top$ and $\widetilde{\gamma}_2(x) = U_4 V_4^\top$ in $n^{1+o(1)}$ time. From Lemma 4.1, the reformulated gradient is $\frac{\mathrm{dLoss}(x)}{\mathrm{d}x} = \widetilde{\mathsf{A}}^\top \mathrm{vec}(\gamma(x))$, and hence the total running time remains $n^{1+o(1)}$. To bound the error, we show that

$$
\begin{aligned}
\left\|\frac{\mathrm{dLoss}(x)}{\mathrm{d}x} - \widetilde{g}\right\|_\infty &= \|\widetilde{\mathsf{A}}^\top (\mathrm{vec}(\gamma(x)) - \mathrm{vec}(\widetilde{\gamma}(x)))\|_\infty \\
&\leq \|\widetilde{\mathsf{A}}^\top\|_\infty \|\gamma(x) - \widetilde{\gamma}(x)\|_\infty \leq \epsilon / \mathrm{poly}(n),
\end{aligned}
$$

This completes the proof. (See full proof at Theorem D.6) □

# 6 HARDNESS

In this section, we provide the lower bound results to compute the gradient of RoPE attention. The hardness result shows that under the widely accepted SETH, the bounded entries condition is necessary for achieving subquadratic runtime.

**Theorem 6.1** (Lower bound, informal version of Theorem E.1). *Assuming* SETH, *for any $q > 0$, for the* $\mathsf{ARAttLGC}(n, d = O(\log n), B = \omega(\sqrt{\log n}))$, *there does not exist an algorithm which can be executed in time $O(n^{2-q})$ based on Def. 3.3.*

*Proof.* See the full proof at Theorem E.1. □

In Theorem 6.1, we show that under the Strong Exponential Time Hypothesis (SETH) (see Hypothesis A.4), computing the gradient of RoPE attention remains computationally hard. Specifically, for any constant $q > 0$, no algorithm can compute the gradient in time $O(n^{2-q})$ when $d = O(\log n)$ and $B = \omega(\sqrt{\log n})$. This result establishes a lower bound that fundamentally limits the efficiency of gradient computation for RoPE attention.

# 7 CONCLUSION

This paper presents the first efficient backward gradient computation, assuming bounded entries for the RoPE-based attention mechanism. We achieve almost linear time complexity by leveraging polynomial methods and the Fast Fourier Transform, making the forward and backward computations comparably efficient. Additionally, we demonstrate that conditions exist under which performance better than quadratic can be realized, consistent with the lower bounds suggested by the Strong Exponential Time Hypothesis (SETH).

These findings not only improve the computational efficiency of RoPE-based attention mechanisms but also provide a foundation for exploring sub-gradient computations in other advanced attention variants of neural networks. This work highlights the connection between algorithm design and computational complexity theory, unveiling new possibilities for the development of large transformer models. Future research could extend these results to cases involving unbounded entries and assess the real-world implications of these theoretical advancements for large language models. Furthermore, applying this approach to other positional encoding mechanisms could further enhance the scalability of state-of-the-art transformer models.

## ETHIC STATEMENT

This paper does not involve human subjects, personally identifiable data, or sensitive applications. We do not foresee direct ethical risks. We follow the ICLR Code of Ethics and affirm that all aspects of this research comply with the principles of fairness, transparency, and integrity.

## REPRODUCIBILITY STATEMENT

We ensure reproducibility of our theoretical results by including all formal assumptions, definitions, and complete proofs in the appendix. The main text states each theorem clearly and refers to the detailed proofs. No external data or software is required.

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

# Appendix

## A  PRELIMINARY

In Section A.1, we talk about polynomial approximation of the exponential function. In Section A.2, we talk about the time complexity of matrix multiplications, setting up the framework for analyzing efficiency in attention computation. In Section A.3, we talk about the Strong Exponential Time Hypothesis (SETH). In Section A.4, we talk about mathematical properties and tricks, such as the tensor trick and row-wise Kronecker products, which enable efficient matrix-vector operations.

### A.1  POLYNOMIAL APPROXIMATION OF EXPONENTIAL

Here, we will explain a technical tool for controlling the error dependence of our approximate algorithm. In particular, we will use the following optimal-degree polynomial approximation of the exponential function.

**Lemma A.1** ((Aggarwal & Alman, 2022)). *Let $B > 1$ and suppose $\epsilon$ in $(0, 0.1)$. We can have $P$, which has input as a scalar and output as a scalar of degree $g$. $g$ is defined as $\Theta\left(\max\left\{\log(1/\epsilon)/(\log(\log(1/\epsilon)/B)), B\right\}\right)$ such that for all $x \in [0, B]$, we can get*

$$|P(x) - \exp(x)| < \epsilon.$$

*Because $P$'s coefficients are rational values with numerators and denominators represented using integers of $\mathrm{poly}(g)$-bit size and these coefficients can be determined in $\mathrm{poly}(g)$ time, we can calculate $P$ in an efficient way.*

### A.2  TIME COMPLEXITY OF MULTIPLICATIONS

Matrix multiplication is a fundamental operation in many algorithms, and understanding its time complexity is essential for analyzing computational efficiency. Here, we introduce the time complexity of matrix multiplications.

**Definition A.2.** *We suppose $n_1, n_2, n_3$, denote any three positive integers. We define $A \in \mathbb{R}^{n_1 \times n_2}$ and $B \in \mathbb{R}^{n_2 \times n_3}$. It costs $\mathcal{T}_{\mathrm{mat}}(n_1, n_2, n_3)$ time to perform $AB$.*

To further analyze the structure of matrix multiplication time complexity, we rely on a well-known fact from prior research (Bürgisser et al., 1997; Bläser, 2013). This fact provides equivalences between different permutations of matrix dimensions.

**Fact A.3.** *We suppose $n_1, n_2, n_3$, denote any three positive integers. $\mathcal{T}_{\mathrm{mat}}(n_1, n_2, n_3) = O(\mathcal{T}_{\mathrm{mat}}(n_1, n_3, n_2)) = O(\mathcal{T}_{\mathrm{mat}}(n_2, n_1, n_3)) = O(\mathcal{T}_{\mathrm{mat}}(n_2, n_3, n_1)) = O(\mathcal{T}_{\mathrm{mat}}(n_3, n_1, n_2)) = O(\mathcal{T}_{\mathrm{mat}}(n_3, n_2, n_1)).$*

### A.3  SETH HYPOTHESIS

Now, we introduce a fundamental theoretical assumption underpinning many of the results presented in this paper: the Strong Exponential Time Hypothesis (SETH). This hypothesis serves as a cornerstone for establishing the hardness of various computational problems.

Our results are built on the common conjecture. (Impagliazzo & Paturi, 2001) introduce the Strong Exponential Time Hypothesis (SETH) as a stronger form of the $P \neq NP$ conjecture. It suggests that our current best SAT algorithms are optimal and is a popular conjecture for proving fine-grained lower bounds for a wide variety of algorithmic problems (Cygan et al., 2016; Williams, 2018).

**Hypothesis A.4** (SETH). *$\forall \epsilon > 0$, $\exists k \in \mathbb{Z}^+$ and $k$ greater or equal to 3 such that, even when utilizing randomized algorithms, within the time of $O(2^{(1-\epsilon)n})$, we cannot solve $k$-SAT problems with $n$ variables.*

### A.4  BASIC FACTS

In this section, we present several basic facts that are used throughout the paper to develop the proof of our main results. These fundamental properties enable efficient computations of vectors and matrices products in our paper.

Here, we first introduce the facts about row-wise Kronecker products.

**Fact A.5** (Row-wise Kronecker product). *Let $U_1, V_1 \in \mathbb{R}^{n \times k_1}$. Let $U_2, V_2 \in \mathbb{R}^{n \times k_2}$. Then we have*

$$(U_1 V_1^\top) \circ (U_2 V_2^\top) = (U_1 \oslash U_2)(V_1 \oslash V_2)^\top$$

*Here, given $U_1 \in \mathbb{R}^{n \times k_1}$ and $U_2 \in \mathbb{R}^{n \times k_2}$, we define the row-wise Kronecker product as $U_1 \oslash U_2 \in \mathbb{R}^{n \times k_1 k_2}$. That is, $(U_1 \oslash U_2)_{i, l_1 + (l_2 - 1)k_1} := (U_1)_{i, l_1} U_{i, l_2}$ for all $i \in [n]$, $l_1 \in [k_1]$ and $l_2 \in [k_2]$*

To simplify the computation of certain matrix operations, we can use a technique known as the tensor trick, which reformulates matrix products into operations involving vectorized representations and Kronecker products.

**Fact A.6** (Tensor trick). *Let $X \in \mathbb{R}^{d \times d}$. Let $x \in \mathbb{R}^{d^2}$ be the vectorization of $X$. Let there be two $n \times d$ matrices $A_1, A_2$, and we define $\mathsf{A} = A_1 \otimes A_2$. Then, we can get $\mathrm{vec}(A_1 X A_2^\top) = \mathsf{A}x$.*

Given the above tensor trick fact, we can derive additional properties that extend its applicability to exponential operations on matrices. These properties can help us compute the exponential of matrix products efficiently. The properties are presented below.

**Fact A.7.** *Let there be two $n \times d$ matrices $A_1, A_2$, and we define $\mathsf{A} = A_1 \otimes A_2$. Let $X \in \mathbb{R}^{d \times d}$. Let $\mathsf{A}_{j_0} \in \mathbb{R}^{n \times d^2}$ be a block of $\mathsf{A}$. We introduce $x \in \mathbb{R}^{d^2}$ as the vectorization of $X$. Thus, we get*

- $(\exp(A_1 X A_2^\top)_{j_0, *})^\top = \exp(\mathsf{A}_{j_0} x)$

- $\mathrm{vec}(\exp(A_1 X A_2^\top)) = \exp(\mathsf{A}x),$

*For the $j_0$-th row of $\exp(A_1 X A_2^\top) \in \mathbb{R}^{n \times n}$, we use the notation $\exp(A_1 X A_2^\top)_{j_0, *}$.*

*Proof.* From Lemma and Def. A.6, we are able to prove this fact. We omit the details here since the proof is straightforward. $\square$

**Fact A.8.** *We suppose there are three vectors of $n$ dimension $x, y, z$. Thus, we get*

- $\langle x \circ y, z \rangle = x^\top \mathrm{diag}(y)z$.

- $\langle x, y \rangle = \langle x \circ y, \mathbf{1}_n \rangle$.

Next, we introduce some important properties of inner products that help us to reshape the equations in the proofs.

**Fact A.9** (Inner Products). *We suppose $n \in \mathbb{Z}^+$, and we suppose the $n$ dimension vectors $a, b, c$ and a scalar $d$. Then, we have*

- $\langle a, b \rangle = \langle a \circ b, \mathbf{1}_n \rangle$.

- $\langle da, b \rangle = d\langle a, b \rangle = \langle a, db \rangle = d\langle b, a \rangle$.

- $\langle a + c, b \rangle = \langle a, b \rangle + \langle c, b \rangle$.

- $\langle a, b \rangle = a^\top b$.

- $\langle a \circ c, b \rangle = \langle a, b \circ c \rangle$.

- $\langle a \circ b, c \rangle = b^\top \mathrm{diag}(a)c$

# B  KEY DEFINITIONS OF ROPE ATTENTION

In this section, we decompose RoPE attention into its individual components, each representing a specific function or operation within the attention mechanism. These definitions provide a structured framework for understanding and analyzing the properties of RoPE attention in subsequent sections.

We denote the $d^4$-dimensional vector $x \in \mathbb{R}^{d^4}$ as the vectorization of a $d^2 \times d^2$ matrix $\mathsf{X}$. We divide the RoPE attention to the following components to simplify our calculations and notation.

First, we define $u(x)$ for the softmax operation.

**Definition B.1** (Softmax $u(x)$)**.** *We suppose there are two $n^2 \times d^2$ matrices $\mathsf{A}, \mathsf{W}$. We define $\widetilde{\mathsf{A}}$ as $\mathsf{A} \oslash \mathsf{W}$, which is an $n^2 \times d^4$ matrix. We use $\widetilde{\mathsf{A}}_{j_0}$ to denote the an $n \times d^4$ subblock of $\widetilde{\mathsf{A}}$, given that the total counts of subblocks is $n$. The function is defined as $u(x)_{j_0}$ maps a $d^4$ dimensional vector to an $n$-dimensional vector with every $j_0 \in [n]$ such that*

$$u(x)_{j_0} := \exp(\widetilde{\mathsf{A}}_{j_0} x).$$

Next, we define $\alpha(x)$ for the diagonal matrix.

**Definition B.2** (Diagonal matrix $\alpha(x)$)**.** *We suppose two $n^2 \times d^2$ matrices $\mathsf{A}, \mathsf{W}$. Suppose that $\widetilde{\mathsf{A}} := \mathsf{A} \oslash \mathsf{W} \in \mathbb{R}^{n^2 \times d^4}$. We use $\widetilde{\mathsf{A}}_{j_0}$ to denote the an $n \times d^4$ subblock of $\widetilde{\mathsf{A}}$, given the counts of total subblocks is $n$. The function is defined as $\alpha(x)_{j_0}$ maps from a $d^4$-dimensional vector to a scalar with every $j_0 \in [n]$ such that*

$$\alpha(x)_{j_0} := \langle \exp(\widetilde{\mathsf{A}}_{j_0} x), \mathbf{1}_n \rangle.$$

We define $s(x)$ for the normalized softmax ($D^{-1} \cdot \text{softmax}$).

**Definition B.3** (Normalized softmax $s(x)$)**.** *From Def. B.1, it defines $u(\cdot)_{j_0}$ , and we have $\alpha(\cdot)_{j_0}$ based on Def. B.2. The function $s(x)_{j_0}$ maps a $d^4$-dimensional vector to an $n$-dimensional vector given every $j_0 \in [n]$ such that $s(x)_{j_0} := \alpha(x)_{j_0}^{-1} u(x)_{j_0}$.*

Lastly, we define $v(y)$ for the value matrix in the attention component.

**Definition B.4** (Value matrix $v(y)$)**.** *Let $A_3 \in \mathbb{R}^{n \times d}$ be a matrix. We define $v(y)_{i_0}$ as the $i_0$-th column of $v(y)$. We define the function $v(y)_{i_0}$ maps a $d^2$-dimensional vector to an $n$-dimensional vector, given each $i_0$ in the set $[d]$, such that $v(y)_{i_0} := A_3 Y_{*,i_0}$ where $y \in \mathbb{R}^{d^2}$ is the vectorization of $n \times n$ matrix $Y$.*

Given the definitions of the RoPE attention components, we can now define the loss functions, which quantify the difference between the computed and target values in the context of RoPE attention.

We first introduce the error $\ell(x)_{j_0,i_0}$ between the exact RoPE attention computation $\langle s(x)_{j_0}, v(y)_{i_0} \rangle$ and approximated RoPE computation $E_{j_0,i_0}$.

**Definition B.5** (RoPE attention error $\ell(x)$)**.** *From Def. B.3, with every $j_0$ in the set $[n]$, it gives $s(x)_{j_0}$ as an $n$-dimensional normalized vector, and we define $v(y)_{i_0}$ based on Def. B.4 given that each $i_0 \in [d]$. Defining a function $\ell(x)_{j_0,i_0}$ maps a $d^4$-dimensional vector to a scalar with each $j_0 \in [n]$ and each $i_0 \in [d]$ such that*

$$\ell(x)_{j_0,i_0} := \langle s(x)_{j_0}, v(y)_{i_0} \rangle - E_{j_0,i_0}.$$

*Here $E_{j_0,i_0}$ is the $(j_0, i_0)$-th coordinate of $E \in \mathbb{R}^{n \times d}$ for each $j_0$ in the set $[n]$ and $i_0$ in the set $[d]$, that is $\ell(x) = s(x)v(y) - E$.*

Then, we define the Loss term.

**Definition B.6** (Loss term $\mathsf{Loss}(x)$)**.** *Here we let $\mathsf{Loss}(x)_{j_0,i_0} := 0.5\ell(x)_{j_0,i_0}^2$ with every $j_0$ in the set $[n]$ and $i_0$ in the set $[d]$.*

## C  RoPE Attention Gradient Calculation

In this section, we analyze the time complexity of exact gradient computation. In Section C.1, we reformulate the closed form of the gradient. In Section C.2, we show the time complexity for $s(x)$ and $v(y)$. In Section C.3, we show the time complexity for $\ell(x)$. In Section C.4, we show the time complexity for $\beta(x)$ and $\gamma(x)$. In Section C.5, we show the total time complexity for computing the gradient of RoPE attention.

In this section, we compute the entry-wise gradient of the RoPE attention loss function from Lemma 3.4

**Lemma C.1.** *If we have for every $i \in [d^4]$,*

- *The column function $u(x)_{j_0} \in \mathbb{R}^n$ (Definitions B.1),*

- $\alpha(x)_{j_0}$ *is a real number (Def. B.2),*

- $s(x)_{j_0}$ *is an arbitrary element in $\mathbb{R}^n$ (Def. B.3),*

- $\ell(x)_{j_0,i_0}$ *is a real number (Def. B.5), and*

- $\mathsf{Loss}(x)_{j_0,i_0}$ *is a real number (Def. B.6).*

*Then, we have $\forall j_0 \in [n], \forall i_0 \in [d]$,*

- **Part 1.**

$$\frac{\mathrm{d}\widetilde{\mathsf{A}}_{j_0}x}{\mathrm{d}x_i} = \underbrace{(\widetilde{\mathsf{A}}_{j_0})_{*,i}}_{n \times 1}.$$

- **Part 2.**

$$\frac{\mathrm{d}u(x)_{j_0}}{\mathrm{d}x_i} = u(x)_{j_0} \circ (\widetilde{\mathsf{A}}_{j_0})_{*,i}.$$

- **Part 3.**

$$\frac{\mathrm{d}\alpha(x)_{j_0}}{\mathrm{d}x_i} = \langle (\widetilde{\mathsf{A}}_{j_0})_{*,i}, u(x)_{j_0} \rangle.$$

- **Part 4.**

$$\frac{\mathrm{d}s(x)_{j_0}}{\mathrm{d}x_i} = -s(x)_{j_0}\langle (\widetilde{\mathsf{A}}_{j_0})_{*,i}, s(x)_{j_0} \rangle + s(x)_{j_0} \circ (\widetilde{\mathsf{A}}_{j_0})_{*,i}$$

- **Part 5.**

$$\frac{\mathrm{d}\langle s(x)_{j_0}, v(y)_{i_0} \rangle}{\mathrm{d}x_i} = \langle -s(x)_{j_0}\langle (\widetilde{\mathsf{A}}_{j_0})_{*,i}, s(x)_{j_0} \rangle + s(x)_{j_0} \circ (\widetilde{\mathsf{A}}_{j_0})_{*,i}, A_3 Y_{*,i_0} \rangle.$$

- **Part 6.**

$$\frac{\mathrm{d}\ell(x)_{j_0,i_0}}{\mathrm{d}x_i} = \langle -s(x)_{j_0}\langle (\widetilde{\mathsf{A}}_{j_0})_{*,i}, s(x)_{j_0} \rangle + s(x)_{j_0} \circ (\widetilde{\mathsf{A}}_{j_0})_{*,i}, A_3 Y_{*,i_0} \rangle.$$

- **Part 7.**

$$\frac{\mathrm{d}\mathsf{Loss}(x)_{j_0,i_0}}{\mathrm{d}x_i} = \ell(x)_{j_0,i_0}\langle -s(x)_{j_0}\langle (\widetilde{\mathsf{A}}_{j_0})_{*,i}, s(x)_{j_0} \rangle + s(x)_{j_0} \circ (\widetilde{\mathsf{A}}_{j_0})_{*,i}, A_3 Y_{*,i_0} \rangle.$$

*Proof.* To show **Part 1**,

$$\begin{aligned}
\frac{\mathrm{d}\widetilde{\mathsf{A}}_{j_0}x}{\mathrm{d}x_i} &= \widetilde{\mathsf{A}}_{j_0}\frac{\mathrm{d}x}{\mathrm{d}x_i} \\
&= \underbrace{\widetilde{\mathsf{A}}_{j_0}}_{n \times d^4}\underbrace{e_i}_{d^4 \times 1} \\
&= (\widetilde{\mathsf{A}}_{j_0})_{*,i},
\end{aligned}$$

and we note that the 1st and 2nd equalities are by the basic derivative rule and the 3rd equality is due to the basis vector definition.

To show **Part 2**,

$$\frac{\mathrm{d}u(x)_{j_0}}{\mathrm{d}x_i} = \frac{\mathrm{d}\exp(\widetilde{\mathsf{A}}_{j_0}x)}{\mathrm{d}x_i}$$

$$= \exp(\widetilde{A}_{j_0} x) \circ \frac{d\widetilde{A}_{j_0} x}{dx_i}$$

$$= \exp(\widetilde{A}_{j_0} x) \circ (\widetilde{A}_{j_0})_{*,i}$$

$$= u(x)_{j_0} \circ (\widetilde{A}_{j_0})_{*,i},$$

and we note that the 1st equality is by Def. B.1, the 2nd equality is by chain rule, the 3rd equality is due to **Part 1**, and the 4th equality is because of Def. B.1.

To show **Part 3**,

$$\frac{d\alpha(x)_{j_0}}{dx_i} = \frac{d\langle \exp(\widetilde{A}_{j_0} x), \mathbf{1}_n \rangle}{dx_i}$$

$$= \langle \frac{d\exp(\widetilde{A}_{j_0} x)}{dx_i}, \mathbf{1}_n \rangle + \langle \exp(\widetilde{A}_{j_0} x), \frac{d\mathbf{1}_n}{dx_i} \rangle$$

$$= \langle \frac{d\exp(\widetilde{A}_{j_0} x)}{dx_i}, \mathbf{1}_n \rangle$$

$$= \langle u(x)_{j_0} \circ (\widetilde{A}_{j_0})_{*,i}, \mathbf{1}_n \rangle$$

$$= \langle (\widetilde{A}_{j_0})_{*,i}, u(x)_{j_0} \rangle,$$

and we note that the 1st equality is by Def. B.2, the 2nd equality is by product rule, the 3rd equality is due to $\frac{d\mathbf{1}_n}{dx_i} = \mathbf{0}_n$, the 4th equality is because of Def. B.1, and 5th equality derives from basic algebra.

To show **Part 4**,

$$\frac{ds(x)_{j_0}}{dx_i} = \frac{d(\alpha(x)_{j_0}^{-1} u(x)_{j_0})}{dx_i}$$

$$= \frac{d\alpha(x)_{j_0}^{-1}}{dx_i} u(x)_{j_0} + \alpha(x)_{j_0}^{-1} \frac{du(x)_{j_0}}{dx_i}$$

$$= -\alpha(x)_{j_0}^{-2} \frac{d\alpha(x)_{j_0}}{dx_i} u(x)_{j_0} + \alpha(x)_{j_0}^{-1} \frac{du(x)_{j_0}}{dx_i}$$

$$= -\alpha(x)_{j_0}^{-2} \langle \underbrace{(\widetilde{A}_{j_0})_{*,i}}_{n \times 1}, \underbrace{u(x)_{j_0}}_{n \times 1} \rangle u(x)_{j_0} + \alpha(x)_{j_0}^{-1} (\underbrace{u(x)_{j_0}}_{n \times 1} \circ \underbrace{(\widetilde{A}_{j_0})_{*,i}}_{n \times 1})$$

$$= -\alpha(x)_{j_0}^{-1} s(x)_{j_0} \langle \underbrace{(\widetilde{A}_{j_0})_{*,i}}_{n \times 1}, \underbrace{u(x)_{j_0}}_{n \times 1} \rangle + s(x)_{j_0} \circ (\widetilde{A}_{j_0})_{*,i}$$

$$= -s(x)_{j_0} \langle \underbrace{(\widetilde{A}_{j_0})_{*,i}}_{n \times 1}, \underbrace{s(x)_{j_0}}_{n \times 1} \rangle + s(x)_{j_0} \circ (\widetilde{A}_{j_0})_{*,i},$$

and we note that the 1st equality is by Def. B.3, the 2nd equality is by product rule, the 3rd equality is due to chain rule, the 4th equality is because of previous parts, the 5th and 6th equalities derive from Def. B.3.

To show **Part 5**,

$$\frac{d\langle s(x)_{j_0}, v(y)_{i_0} \rangle}{dx_i} = \langle \frac{ds(x)_{j_0}}{dx_i}, v(y)_{i_0} \rangle + \langle s(x)_{j_0}, \frac{dv(y)_{i_0}}{dx_i} \rangle$$

$$= \langle \frac{ds(x)_{j_0}}{dx_i}, v(y)_{i_0} \rangle$$

$$= \langle -s(x)_{j_0} \langle (\widetilde{A}_{j_0})_{*,i}, s(x)_{j_0} \rangle + s(x)_{j_0} \circ (\widetilde{A}_{j_0})_{*,i}, A_3 Y_{*,i_0} \rangle,$$

and we note that the 1st equality is due to the product rule, the 2nd equality is by $\frac{dv(y)_{i_0}}{dx_i} = \mathbf{0}_n$, and the 3rd equality is due to the previous part.

To show **Part 6**,

$$\frac{d\ell(x)_{j_0,i_0}}{dx_i} = \frac{d(\langle s(x)_{j_0}, v(y)_{i_0} \rangle - E_{j_0,i_0})}{dx_i}$$

$$= \frac{\mathrm{d}\langle s(x)_{j_0}, v(y)_{i_0}\rangle}{\mathrm{d}x_i}$$

$$= \langle -s(x)_{j_0} \underbrace{\langle (\widetilde{\mathsf{A}}_{j_0})_{*,i}, \underbrace{s(x)_{j_0}}_{n \times 1} \rangle}_{n \times 1} + s(x)_{j_0} \circ (\widetilde{\mathsf{A}}_{j_0})_{*,i}, A_3 Y_{*,i_0}\rangle,$$

and we note that the 1st equality is by Def. B.5, the 2nd equality is by $\frac{\mathrm{d}E_{j_0,i_0}}{\mathrm{d}x_i} = \mathbf{0}_n$, and the 3rd equality is due to the previous part.

To show **Part 7**,

$$\frac{\mathrm{d}\mathsf{Loss}(x)_{j_0,i_0}}{\mathrm{d}x_i} = 0.5\frac{\mathrm{d}(\ell(x)_{j_0,i_0})^2}{\mathrm{d}x_i}$$

$$= \ell(x)_{j_0,i_0} \cdot \frac{\mathrm{d}\ell(x)_{j_0,i_0}}{\mathrm{d}x_i}$$

$$= \ell(x)_{j_0,i_0}\langle -s(x)_{j_0}\underbrace{\langle (\widetilde{\mathsf{A}}_{j_0})_{*,i}, \underbrace{s(x)_{j_0}}_{n \times 1}\rangle}_{n \times 1} + s(x)_{j_0} \circ (\widetilde{\mathsf{A}}_{j_0})_{*,i}, A_3 Y_{*,i_0}\rangle,$$

and we note that the 1st equality is by Def. B.6, the 2nd equality is by chain rule and the 3rd equality is due to the previous part. $\qquad\square$

## C.1 REFORMULATE THE GRADIENT INTO ITS CLOSED FORM

In this section, we reformulate the entry-wise gradient of the RoPE loss function from Lemma 4.1 into its matrix form.

We first begin with reformulating the gradient with respect to the entire vector $x$.

**Lemma C.2** (Gradient Reformulation, $\frac{\mathrm{d}\mathsf{Loss}(x)_{j_0,i_0}}{\mathrm{d}x}$). *If we have for every $i \in [d^4]$,*

- *The column function $u(x)_{j_0} \in \mathbb{R}^n$ (Definitions B.1),*

- *$\alpha(x)_{j_0}$ is a real number (Def. B.2),*

- *$s(x)_{j_0}$ is an arbitrary element in $\mathbb{R}^n$ (Def. B.3),*

- *$\ell(x)_{j_0,i_0}$ is a real number (Def. B.5), and*

- *$\mathsf{Loss}(x)_{j_0,i_0}$ is a real number (Def. B.6).*

*Then, we have*

$$\frac{\mathrm{d}\mathsf{Loss}(x)_{j_0,i_0}}{\mathrm{d}x} = \ell(x)_{j_0,i_0}\widetilde{\mathsf{A}}_{j_0}^{\top}(\mathrm{diag}(s(x)_{j_0})A_3 Y_{*,i_0} - s(x)_{j_0}s(x)_{j_0}^{\top}A_3 Y_{*,i_0}).$$

*Proof.*

$$\frac{\mathrm{d}\mathsf{Loss}(x)_{j_0,i_0}}{\mathrm{d}x_i} = \ell(x)_{j_0,i_0}\langle -s(x)_{j_0}\langle (\widetilde{\mathsf{A}}_{j_0})_{*,i}, s(x)_{j_0}\rangle + s(x)_{j_0} \circ (\widetilde{\mathsf{A}}_{j_0})_{*,i}, A_3 Y_{*,i_0}\rangle$$

$$= \ell(x)_{j_0,i_0}\langle s(x)_{j_0} \circ (\widetilde{\mathsf{A}}_{j_0})_{*,i}, A_3 Y_{*,i_0}\rangle - \ell(x)_{j_0,i_0}\langle s(x)_{j_0}\langle (\widetilde{\mathsf{A}}_{j_0})_{*,i}, s(x)_{j_0}\rangle, A_3 Y_{*,i_0}\rangle$$

$$= \ell(x)_{j_0,i_0}\langle s(x)_{j_0} \circ (\widetilde{\mathsf{A}}_{j_0})_{*,i}, A_3 Y_{*,i_0}\rangle - \ell(x)_{j_0,i_0}\langle (\widetilde{\mathsf{A}}_{j_0})_{*,i}, s(x)_{j_0}\rangle\langle s(x)_{j_0}, A_3 Y_{*,i_0}\rangle$$

$$= \ell(x)_{j_0,i_0}(\widetilde{\mathsf{A}}_{j_0}^{\top})_{*,i}\mathrm{diag}(s(x)_{j_0})A_3 Y_{*,i_0} - \ell(x)_{j_0,i_0}(\widetilde{\mathsf{A}}_{j_0}^{\top})_{*,i}s(x)_{j_0}s(x)_{j_0}^{\top}A_3 Y_{*,i_0}$$

$$= \ell(x)_{j_0,i_0}(\widetilde{\mathsf{A}}_{j_0}^{\top})_{*,i}(\mathrm{diag}(s(x)_{j_0}) - s(x)_{j_0}s(x)_{j_0}^{\top})A_3 Y_{*,i_0}.$$

where the 1st step follows from Lemma 4.1, and all other steps follow from Fact A.9.

Then, the gradient can be reformulated as follows.

$$\frac{\mathrm{d}\mathsf{Loss}(x)_{j_0,i_0}}{\mathrm{d}x} = \ell(x)_{j_0,i_0}\widetilde{\mathsf{A}}_{j_0}^{\top}(\mathrm{diag}(s(x)_{j_0})A_3 Y_{*,i_0} - s(x)_{j_0}s(x)_{j_0}^{\top}A_3 Y_{*,i_0}).$$

Thus, we complete the proof. $\qquad\square$

Next, we show our reformulation of the gradient by dropping the index $i_0$ from $\mathsf{Loss}(x)_{j_0,i_0}$

**Lemma C.3** (Gradient Reformulation, $\frac{\mathrm{d}\mathsf{Loss}(x)_{j_0}}{\mathrm{d}x}$). *If we have for every $i \in [d^4]$,*

- *The column function $u(x)_{j_0} \in \mathbb{R}^n$ (Definitions B.1),*

- *$\alpha(x)_{j_0}$ is a real number (Def. B.2),*

- *$s(x)_{j_0}$ is an arbitrary element in $\mathbb{R}^n$ (Def. B.3),*

- *$\ell(x)_{j_0,i_0}$ is a real number (Def. B.5),*

- *$\mathsf{Loss}(x)_{j_0,i_0}$ is a real number (Def. B.6), and*

- *$\beta(x)_{j_0} \in \mathbb{R}^n$ is $\underbrace{A_3 Y}_{n \times d} \underbrace{\ell(x)_{j_0,*}^\top}_{d \times 1}$.*

*Then, we get*

$$
\frac{\mathrm{d}\mathsf{Loss}(x)_{j_0}}{\mathrm{d}x} = \widetilde{\mathsf{A}}_{j_0}^\top (s(x)_{j_0} \circ A_3 \beta(x)_{j_0})
$$
$$
- \widetilde{\mathsf{A}}_{j_0}^\top s(x)_{j_0} \langle s(x)_{j_0}, A_3 \beta(x)_{j_0} \rangle.
$$

*Proof.* We can get

$$
\frac{\mathrm{d}\mathsf{Loss}(x)_{j_0}}{\mathrm{d}x}
$$
$$
= \sum_{i_0 \in [d]} \frac{\mathrm{d}\mathsf{Loss}(x)_{j_0,i_0}}{\mathrm{d}x}
$$
$$
= \sum_{i_0 \in [d]} \widetilde{\mathsf{A}}_{j_0}^\top (\mathrm{diag}(s(x)_{j_0}) - s(x)_{j_0} s(x)_{j_0}^\top) \ell(x)_{j_0,i_0} A_3 Y_{*,i_0}
$$
$$
= \widetilde{\mathsf{A}}_{j_0}^\top (\mathrm{diag}(s(x)_{j_0}) - s(x)_{j_0} s(x)_{j_0}^\top) \beta(x)_{j_0},
$$

and we note that the first equality is because Lemma 3.4, the 2nd equality is due to basic algebra, and the 3rd equality comes from the lemma statement.

Thus, we complete this proof. $\square$

Finally, we reformulate the gradient into its matrix form.

**Lemma C.4** (Gradient Reformulation, $\frac{\mathrm{d}\mathsf{Loss}(x)}{\mathrm{d}x}$, Formal version of Lemma 4.1). *If we have for every $i \in [d^4]$,*

- *The column function $u(x)_{j_0} \in \mathbb{R}^n$ (Definitions B.1),*

- *$\alpha(x)_{j_0}$ is a real number (Def. B.2),*

- *$s(x)_{j_0}$ is an arbitrary element in $\mathbb{R}^n$ (Def. B.3),*

- *$\ell(x)_{j_0,i_0}$ is a real number (Def. B.5),*

- *$\mathsf{Loss}(x)_{j_0,i_0}$ is a real number (Def. B.6),*

- *$\beta(x)_{j_0} \in \mathbb{R}^n$ is $\underbrace{A_3 Y}_{n \times d} \underbrace{\ell(x)_{j_0,*}^\top}_{d \times 1}$, and*

- *$\gamma(x)_{j_0} \in \mathbb{R}^n$ is $(\mathrm{diag}(s(x)_{j_0}) - s(x)_{j_0} s(x)_{j_0}^\top) \beta(x)_{j_0}$*

*Then, we get*

$$
\frac{\mathrm{d}\mathsf{Loss}(x)}{\mathrm{d}x} = \underbrace{\widetilde{\mathsf{A}}^\top}_{d^4 \times n^2} \mathrm{vec}(\underbrace{\gamma(x)}_{n \times n})
$$

*Proof.* We show that

$$\frac{\mathrm{dLoss}(x)}{\mathrm{d}x} = \sum_{j_0 \in [n]} \frac{\mathrm{dLoss}(x)_{j_0}}{\mathrm{d}x}$$

$$= \sum_{j_0 \in [n]} \widetilde{\mathsf{A}}_{j_0}^\top \gamma(x)_{j_0}$$

$$= \widetilde{\mathsf{A}}^\top \mathrm{vec}(\gamma(x)),$$

where we note that the 1st equality is because of Lemma 3.4, the 2nd equality is based on the lemma statement, and the 3rd equality derives from basic concepts of vectorization.

Thus, we complete the proof. $\qquad\square$

## C.2 TIME COMPLEXITY FOR COMPUTING $s(x)$ AND $v(y)$ FUNCTIONS

In this section, we use the vector and matrix multiplication time complexity from Definition A.2 and Fact A.3 to analyze the complexity of computing $s(x)$ and $v(y)$.

**Lemma C.5.** *Pick $s(x)$ and $v(y)$ from Def. B.3 and Def. B.4, then it costs $O(\mathcal{T}_{\mathrm{mat}}(n,d,d) + \mathcal{T}_{\mathrm{mat}}(n,d,n))$ time to get $s(x)$, and it costs $\mathcal{T}_{\mathrm{mat}}(n,d,d)$ time to get $v(y)$.*

*Proof.* We first show the time complexity of $s(x)$.

Let $A \in \mathbb{R}^{n \times n}$ be the RoPE attention matrix. Let $D = A\mathbf{1}_n$. Then

$$s(x) = D^{-1}A.$$

Then, we need $\mathcal{T}_{\mathrm{mat}}(n,d,d) + \mathcal{T}_{\mathrm{mat}}(n,d,n)$ time to get $A$.

Next, we need $O(n^2)$ time to get $D$.

Now, we need $O(n^2)$ time to get $D^{-1}A$.

Therefore, they cost time $O(\mathcal{T}_{\mathrm{mat}}(n,d,d) + \mathcal{T}_{\mathrm{mat}}(n,d,n))$ time .

We show the time complexity of $v(y)$.

To get $v(y) = A_3 Y$, it costs time $\mathcal{T}_{\mathrm{mat}}(n,d,d)$.

Thus, we complete the proof. $\qquad\square$

## C.3 TIME COMPLEXITY FOR COMPUTING $\ell(x)$ FUNCTIONS

In this section, we use the vector and matrix multiplication time complexity from Definition A.2 and Fact A.3 to analyze the complexity of computing $\ell(x)$.

**Lemma C.6.** *We have $\ell(x)$ from Def. B.5, then it costs $\mathcal{T}_{\mathrm{mat}}(n,n,d) + O(nd)$ to calculate $\ell(x)$.*

*Proof.* We show the time complexity of $\ell(x)$, where $\ell(x) = s(x)v(y) - E$.

It costs $\mathcal{T}_{\mathrm{mat}}(n,n,d)$ time to get $s(x)v(y)$.

Then, it requires $O(nd)$ time to get $s(x)v(y) - E$.

Therefore, they cost time $\mathcal{T}_{\mathrm{mat}}(n,n,d) + O(nd)$.

Thus, we complete the proof. $\qquad\square$

## C.4 TIME COMPLEXITY FOR COMPUTING $\beta(x)$ AND $\gamma(x)$ FUNCTIONS

In this section, we use the vector and matrix multiplication time complexity from Definition A.2 and Fact A.3 to analyze the complexity of computing $\beta(x)$ and $\gamma(x)$.

**Lemma C.7.** *Let $\beta(x) \in \mathbb{R}^{n \times n}$ be defined as $\beta(x) := \ell(x)v(y)^\top$ and $\gamma(x)$ be defined as $\gamma(x)_{j_0} := (\mathrm{diag}(s(x)_{j_0} - s(x)_{j_0}s(x)_{j_0}^\top \beta(x)_{j_0} \in \mathbb{R}^n$, given that $s(x) \in \mathbb{R}^{n \times n}$ then $\beta(x)$ can be computed in time of $O(\mathcal{T}_{\mathrm{mat}}(n, n, d))$ and $\gamma(x)$ can be computed in time of $O(n^2)$.*

*Proof.* Here we present $\beta(x)$ as follows: $\beta(x) = \ell(x)v(y)^\top$.

It costs $\mathcal{T}_{\mathrm{mat}}(n, d, n)$ time to get $\ell(x)v(y)^\top$, which equals to $O(\mathcal{T}_{\mathrm{mat}}(n, n, d))$.

Next, we show the time complexity for $\gamma(x)_{j_0} = (\mathrm{diag}(s(x)_{j_0} - s(x)_{j_0}s(x)_{j_0}^\top)\beta(x)_{j_0}$ It costs $O(n)$ time to get $\gamma(x)_{j_0}$. The reason is that $s(x)_{j_0}s(x)_{j_0}^\top$ is a rank one matrix and $\mathrm{diag}(s(x)_{j_0})$ is a diagonal matrix

Given $j_0 \in [n]$, the time for $\gamma(x)$ is $O(n^2)$ and we finish the proof. $\qquad\square$

### C.5 Time Complexity for Computing the Gradient of RoPE Attention

**Theorem C.8** (RoPE attention gradient computation time complexity, Restatement of Theorem 4.2)**.**
*We define three $n \times d$ input sequence matrices as $A_1, A_2, A_3$, and the $n \times d$ approximated attention computation matrix as $E$. We define several input fixed matrices as $X_1, X_2, Y \in \mathbb{R}^{d \times d}$. We define $\mathsf{X} = X_1 \otimes X_2$, $\mathsf{A} = A_1 \otimes A_2$. We define $x := \mathrm{vec}(\mathsf{X})$ and try to get the Loss function gradient. Let $g := \frac{d\mathrm{Loss}(X_1, X_2)}{dx}$ where $\mathrm{Loss}(X_1, X_2)$ from Def. 3.3. Then, it costs $O(\mathcal{T}_{\mathrm{mat}}(n, d, d) + \mathcal{T}_{\mathrm{mat}}(n, d, n))$ time to get the gradient $g \in \mathbb{R}^{d^4}$.*

*Proof.* We show the time complexity of $g$ as follows.

1. We need time $O(\mathcal{T}_{\mathrm{mat}}(n, d, d) + \mathcal{T}_{\mathrm{mat}}(n, d, n))$ for $s(x), v(y)$ from Lemma C.2.

2. We need time $O(\mathcal{T}_{\mathrm{mat}}(n, n, d) + \mathcal{T}_{\mathrm{mat}}(n, d, d))$ for $\ell(x)$ from Lemma C.3.

3. We need time $O(\mathcal{T}_{\mathrm{mat}}(n, n, d))$ for $\beta(x)$ from Lemma C.7.

4. We need time $O(n^2)$ for $\gamma(x)$ from Lemma C.7.

Therefore, it costs $O(\mathcal{T}_{\mathrm{mat}}(n, d, d) + \mathcal{T}_{\mathrm{mat}}(n, d, n))$ time overall for the gradient computation.

Thus, we complete the proof. $\qquad\square$

## D  Low Rank Approximation of RoPE Attention

This section presents the fast running time using the low-rank approximations where the low-rank matrices are generated from the polynomial method (see Lemma A.1).

### D.1 Approximate $s$ Using Low Rank Approximation

In this section, we use the low-rank approximation technique to approximate $s(x)$

**Lemma D.1** (Low Rank Approximate $s(x)$)**.** *For any $B = o(\sqrt{\log n})$, let $k_1$ equals to $n^{o(1)}$ such that: Suppose we have two $n \times d$ matrices $A_1, A_2$, $X_1, X_2 \in \mathbb{R}^{d \times d}$ and $\mathsf{X} = X_1 \otimes X_2 \in \mathbb{R}^{d^2 \times d^2}$. Assume we can use $O(\log n)$ bits to write every entry from $s(x)$. It holds that $\max\{\|A_1X_1\|_\infty, \|A_2X_2\|_\infty\} \leq B$, then there are three matrices $U_1, V_1, W_1 \in \mathbb{R}^{n \times k_1}$ such that $\|U_1V_1^\top - s(x)\|_\infty \leq \epsilon/\mathrm{poly}(n)$. Here $s(x) = D^{-1}A \in \mathbb{R}^{n \times n}$ where $A$ is defined as the matrix representation of $\exp((\mathsf{A} \oslash \mathsf{W})\mathrm{vec}(X))$, and $D = \mathrm{diag}(A/d)\mathbf{1}_n$. Moreover, these matrices $U_1, V_1$ can be created explicitly in $n^{1+o(1)}$ time.*

*Proof.* By definition of $A(\mathsf{X})$, we have

$$\mathrm{vec}(A(\mathsf{X})) = \exp(\mathsf{A} \oslash \mathsf{W})\mathrm{vec}(X).$$

Hence, using the tensor trick, we have

$$A(\mathsf{X})_{i,j} = \exp((\mathsf{A}_{i+(j-1)n} \otimes \mathsf{W}_{i+(j-1)n}) \operatorname{vec}(\mathsf{X})/d)$$
$$= \exp((\mathsf{A}_{i+(j-1)n} \otimes w_{i-j}^\top) \operatorname{vec}(\mathsf{X})/d).$$

We define $w_{i-j} := \operatorname{vec}(W_{i-j}) \in \mathbb{R}^{d^2}$ and define $\mathsf{W}$ such that $\mathsf{W}_{j_0}$ is an $1 \times d^2$ block and $\mathsf{W}_{i+(j-1)n} := w_{i-j}^\top$. We also define $\mathsf{A} := A_1 \otimes A_2 \in \mathbb{R}^{n^2 \times d^2}$ and $\mathsf{X} := X_1 \otimes X_2 \in \mathbb{R}^{d^2 \times d^2}$. We use $\mathsf{A}_{j_0}$ to denote the a $1 \times d^2$ subblock of $\mathsf{A}$.

We can reformulate the attention matrix $A$ as, for $i, j \in [n]$

$$A(\mathsf{X})_{i,j} = \exp(\underbrace{\mathsf{A}_{i+(j-1)n}}_{1 \times d^2} \underbrace{\mathsf{X}}_{d^2 \times d^2} \underbrace{w_{i-j}/d}_{d^2 \times 1}).$$

Thus, we can show that

$$\mathsf{A}_{i+(j-1)n}\mathsf{X}w_{i-j},$$
$$= (A_{1,i,*} \otimes A_{2,j,*})(X_1 \otimes X_2) \operatorname{vec}(W_{i-j})$$
$$= A_{1,i,*} X_1 W_{i-j} X_2^\top A_{2,j,*}^\top$$

where 1st equality uses definitions of $w_{i-j}, \mathsf{A}$, and $\mathsf{X}$, and the second step uses the tensor trick. We complete our proof after applying Lemma 5.1. $\square$

### D.2 APPROXIMATE $\ell$ USING LOW RANK APPROXIMATION

In this section, we use the low-rank approximation technique to approximate $\ell(x)$

**Lemma D.2** (Low Rank Approximate $\ell(x)$). *Let $d$ equal $O(\log n)$. Suppose we can use $O(\log n)$ bits to write every entry in $E, v(y) \in \mathbb{R}^{n \times d}$. Define the $\ell(x) \in \mathbb{R}^{n \times d}$ as specified in Def. B.5. Then, we have $U_1, V_1 \in \mathbb{R}^{n \times k_1}$ such that $\|U_1 V_1^\top v(y) - E - \ell(x)\|_\infty \leq \epsilon/\operatorname{poly}(n)$.*

*Proof.* Here, we present the bound as follows.

$$\|U_1 V_1^\top v(y) - E - \ell(x)\|_\infty = \|U_1 V_1^\top v(y) - s(x)v(y)\|_\infty$$
$$= \|v(y)\|_\infty \cdot \|U_1 V_1^\top - s(x)\|_\infty$$
$$\leq \epsilon/\operatorname{poly}(n),$$

where the 1st is because of Def. B.5, 2nd step is based on the distributive law, and 3rd step is due to Lemma D.1. $\square$

### D.3 APPROXIMATE $\beta$ USING LOW RANK APPROXIMATION

In this section, we use the low-rank approximation technique to approximate $\beta(x)$

**Lemma D.3** (Low Rank Approximate $\beta(x)$). *Let $k_2 = n^{o(1)}$. We define $\ell(x) \in \mathbb{R}^{n \times d}$ based on Def. B.5, and $v(y) \in \mathbb{R}^{n \times d}$ based on Def. B.4. We suppose $\beta(x)$ is equal to $v(y)\ell(x)^\top$, which is an $n \times n$ matrix. Let $U_2, V_2 \in \mathbb{R}^{n \times k_2}$ such that $\|U_2 V_2^\top - \beta(x)\|_\infty \leq \epsilon/\operatorname{poly}(n)$. In $n^{1+o(1)}$ time, we can get $U_2, V_2$.*

*Proof.* Let $\widetilde{\beta}(x) \approx \beta(x)$

By Lemma D.2, $U_1 V_1^\top v(y) - E$ approximately equals to $\ell(x)$.

Then we define $\widetilde{\beta}(x) = v(y)(U_1 V_1^\top v(y) - E)^\top$.

We can use the low-rank technique to represent $\widetilde{\beta}(x) = v(y)v(y)^\top V_1 U_1^\top - v(y)E^\top$.

Also, $v(y)^\top V_1$ can be computed at first because it takes $n^{1+o(1)}$ time.

Given that all low-rank matrices, we have $U_2, V_2 \in \mathbb{R}^{n \times k_2}$ where $k_2 = \max\{d, k\} + d = n^{o(1)}$.

Here, we present the proof for obtaining the bound.

$$\|\widetilde{\beta}(x) - \beta(x)\|_\infty = \|v(y)(U_1 V_1^\top v(y)) - E)^\top - v(y)\ell(x)^\top\|_\infty$$
$$\leq \|U_1 V_1^\top v(y)) - E - \ell(x)\|_\infty \cdot \|v(y)\|_\infty \cdot d$$
$$\leq \epsilon / \operatorname{poly}(n)$$

where the first step is based on the definition of $\beta(x)$ and $\widetilde{\beta}(x)$, the second step is due to the distributive law, and the third step derives from Lemma D.2.

Thus, we complete the proof. $\qquad\square$

## D.4 APPROXIMATE $\gamma$ USING LOW RANK APPROXIMATION

In this section, we use the low-rank approximation technique to approximate $\gamma(x)$. Specifically, we apply the polynomial methods to $\gamma_1(x)$ and $\gamma_2(x)$ where $\gamma(x) = \gamma_1(x) - \gamma_2(x)$.

First, we show the low-rank approximation of $\gamma_1(x)$.

**Lemma D.4** (Low Rank Approximate $\gamma_1(x)$). *Let $k_1 = n^{o(1)}$. Let $k_2 = n^{o(1)}$. We suppose $\gamma_1(x)$ is $\operatorname{diag}(s(x))\beta(x)$, and $U_1, V_1$ be two $n \times k_1$ matrices, in which $\|U_1 V_1^\top - f(x)\|_\infty \leq \frac{\epsilon}{\operatorname{poly}(n)}$. We suppose two $n \times k_2$ matrices $U_2, V_2$ in which $\|U_2 V_2^\top - \beta(x)\|_\infty \leq \frac{\epsilon}{\operatorname{poly}(n)}$. Then we have two $n \times k_3$ matrices in which $\|U_3 V_3^\top - \gamma_1(x)\|_\infty \leq \epsilon / \operatorname{poly}(n)$. We can construct $U_3, V_3$ in $n^{1+o(1)}$ time.*

*Proof.* Let $U_3 = U_1 \oslash U_2$ and $V_3 = V_1 \oslash V_2$, and we can use $n^{1+o(1)}$ time to get them.

Let $\widetilde{s}(x) = U_1 V_1^\top$ and $\widetilde{\beta}(x) = U_2 V_2^\top$.

Then, we have the following by Fact A.5.

$$\|U_3 V_3^\top - \gamma_1(x)\|_\infty \leq \|U_3 V_3^\top - \operatorname{diag}(s(x))\beta(x)\|_\infty$$
$$= \|(U_1 \oslash U_2)(V_1 \oslash V_2)^\top - \operatorname{diag}(s(x))\beta(x)\|_\infty$$
$$= \|\operatorname{diag}(U_1 V_1^\top)(U_2 V_2^\top) - \operatorname{diag}(s(x))\beta(x)\|_\infty$$
$$= \|\operatorname{diag}(\widetilde{s}(x))\widetilde{\beta}(x) - \operatorname{diag}(s(x))\beta(x)\|_\infty$$
$$= \|\operatorname{diag}(\widetilde{s}(x))\widetilde{\beta}(x) - \operatorname{diag}(\widetilde{s}(x))\beta(x) + \operatorname{diag}(\widetilde{s}(x))\beta(x) - \operatorname{diag}(s(x))\beta(x)\|_\infty$$
$$\leq \|\operatorname{diag}(\widetilde{s}(x))\widetilde{\beta}(x) - \operatorname{diag}(\widetilde{s}(x))\beta(x)\|_\infty + \|\operatorname{diag}(\widetilde{s}(x))\beta(x) - \operatorname{diag}(s(x))\beta(x)\|_\infty$$
$$\leq \frac{\epsilon}{\operatorname{poly}(n)}$$

where the first inequality is because of the def. of $\gamma_1(x)$, the second equality is due to the def. of $U_3, V_3$, the third equality is based on Fact A.5, the fourth equality is due to the def. of $\widetilde{s}(x)$ and $\widetilde{\beta}(x)$, the fifth equality is due to simple arithmetic, the sixth inequality is because of the triangle inequality, and the seventh inequality derives from Lemma D.1 and Lemma D.3. $\qquad\square$

Next, we show the low-rank approximation of $\gamma_2(x)$.

**Lemma D.5** (Low Rank Approximate $\gamma_2(x)$). *Let $k_1 = n^{o(1)}$. Let $k_2 = n^{o(1)}$. Let $k_4 = n^{o(1)}$. Let $\gamma_2(x) \in \mathbb{R}^{n \times n}$ where for $j_0$ in set $[n]$, $j_0$ represents $j_0$-th column, $\gamma_2(x)_{j_0} = s(x)_{j_0} s(x)_{j_0}^\top \beta(x)_{j_0}$. We suppose $U_1, V_1 \in \mathbb{R}^{n \times k_1}$ in which $\|U_1 V_1^\top - s(x)\|_\infty \leq \frac{\epsilon}{\operatorname{poly}(n)}$. We suppose two $n \times k_2$ matrices $U_2, V_2$ in which $\|U_2 V_2^\top - \beta(x)\|_\infty \leq \frac{\epsilon}{\operatorname{poly}(n)}$. Then, we have $U_4, V_4 \in \mathbb{R}^{n \times k_4}$ such that $\|U_4 V_4^\top - \gamma_2(x)\|_\infty \leq \epsilon / \operatorname{poly}(n)$. We can get $U_4, V_4$ in $n^{1+o(1)}$ time.*

*Proof.* Let $\rho(x) \in \mathbb{R}^n$ be $\rho(x)_{j_0} := s(x)_{j_0} \beta(x)_{j_0}$.

We define $\widetilde{\rho}(x) \approx \rho(x)$.

Let $(U_1 V_1)_{j_0,*}^\top \approx s(x)_{j_0}$ and $(U_2 V_2)_{j_0,*}^\top \approx \beta(x)_{j_0}$.

Then, we define $\widetilde{\rho}(x)_{j_0}$ as the inner product of $\widetilde{s}(x)_{j_0}$ and $\widetilde{\beta}(x)_{j_0}$, and by Fact A.9, we have $\widetilde{\rho}(x)_{j_0} = (U_1 V_1)_{j_0,*} \cdot (U_2 V_2)_{j_0,*}^\top$

Then, it costs $n^{1+o(1)}$ time if we compute $V_1 V_2^\top$ first.

Now, we show

$$\widetilde{\rho}(x)_{j_0} = (U_1 V_1)_{j_0,*} \cdot (U_2 V_2)_{j_0,*}^\top$$
$$= \underbrace{(U_1)_{j_0,*}}_{1 \times k_1} \underbrace{V_1 V_2^\top}_{k_1 \times k_2} \underbrace{(U_2)_{j_0,*}^\top}_{k_2 \times 1}$$

Once the $V_1 V_2^\top$ are pre-computed, the above step only takes $O(k_1 k_2)$ time. Given that $j_0 \in [n]$, we can have the total time $O(n k_1 k_2) = n^{1+o(1)}$.

We suppose $\widetilde{s}(x)$ approximates $s(x)$ and set is equal to $U_1 V_1^\top$. Then, we are able to approximate $\gamma_2(x)$ using $\widetilde{s}(x)$ and $\widetilde{\rho}(x)$ as follows.

We suppose $\widetilde{\gamma}_2(x)$ equals to $\widetilde{s}(x) \operatorname{diag}(\widetilde{\rho}(x))$. $U_4$ and $V_4$ can be obtained since we can use the low-rank approximation technique to represent $\widetilde{s}(x)$ and $\operatorname{diag}(\widetilde{\rho}(x))$ is a diagonal matrix. Basically $U_4 = U_1$ and $V_4 = \operatorname{diag}(\widetilde{\rho}(x)) V_1$.

Now, we need to control the error. We have

$$\|U_4 V_4^\top - \gamma_2(x)\|_\infty = \|\widetilde{\gamma}_2(x) - \gamma_2(x)\|_\infty$$
$$= \max_{j_0 \in [n]} \|\widetilde{s}(x)_{j_0} \widetilde{\rho}(x)_{j_0} - s(x)_{j_0} \rho(x)_{j_0}\|_\infty$$
$$= \max_{j_0 \in [n]} \|\widetilde{s}(x)_{j_0} \widetilde{\rho}(x)_{j_0} - \widetilde{s}(x)_{j_0} \rho(x)_{j_0} + \widetilde{s}(x)_{j_0} \rho(x)_{j_0} - s(x)_{j_0} \rho(x)_{j_0}\|_\infty$$
$$\leq \max_{j_0 \in [n]} \|\widetilde{s}(x)_{j_0} \widetilde{\rho}(x)_{j_0} - \widetilde{s}(x)_{j_0} \rho(x)_{j_0}\|_\infty + \|\widetilde{s}(x)_{j_0} \rho(x)_{j_0} - s(x)_{j_0} \rho(x)_{j_0}\|_\infty$$
$$\leq \max_{j_0 \in [n]} \|\widetilde{s}(x)_{j_0}\|_\infty \cdot \|\widetilde{\rho}(x)_{j_0} - \rho(x)_{j_0}\|_\infty + \|\widetilde{s}(x)_{j_0} - s(x)_{j_0}\|_\infty \cdot \|\rho(x)_{j_0}\|_\infty$$
$$\leq \epsilon / \operatorname{poly}(n)$$

where the 1st equality is based on the def. of $\widetilde{\gamma}_2(x)$, the 2nd equality is due to def. of $\widetilde{\gamma}_2(x)$ and $\gamma_2(x)$, the 3rd equality is due to simple mathematical properties, the 4th step is due to the triangle inequalities, and the 5th step is due to the distributive law.

Thus, we complete the proof. $\square$

## D.5  FAST COMPUTATION IN ALMOST LINEAR TIME

In this section, we present our main result. With the low-rank approximation, we can approximate the RoPE gradient computations in almost linear time.

**Theorem D.6** (Main result, Low Rank Approximate RoPE Attention Gradient, Restatement of Theorem 5.7)**.** *Assuming the entries of $A_1, A_2, X_1, X_2, Y, E$ are represented using $O(\log n)$ bits, there is an $n^{1+o(1)}$ time algorithm to solve* AAttLGC$(n, d = O(\log n), B = o(\sqrt{\log n}))$*, from Def. 3.3, with the accuracy upper bounded by $\frac{1}{\operatorname{poly}(n)}$ . To be more specific, a gradient vector $\widetilde{g} \in \mathbb{R}^{d^4}$ comes out of our algorithm where $\|\frac{d\mathsf{Loss}}{dx} - \widetilde{g}\|_\infty \leq \frac{1}{\operatorname{poly}(n)}$.*

*Proof.* By Lemma D.5 and Lemma D.4, There are matrices $\gamma(x), \gamma_1(x) \in \mathbb{R}^{n \times n}$ and $\gamma_2(x)$, we have

$$\gamma(x) = \gamma_1(x) - \gamma_2(x).$$

We assume Lemma D.4 and Lemma D.5 are true from Lemma D.1 to Lemma D.3. Thus, we can have the following based on Lemma D.4 Lemma D.5.

We can use low-rank approximation technique to represent $\widetilde{\gamma}_1(x) = U_3 V_3^\top$ and $\widetilde{\gamma}_2(x) = U_3 V_3^\top$ as the approximation to $\gamma_1(x)$ and $\gamma_2(x)$ respectively.

The cost is $n^{1+o(1)}$ time for every Lemma in Lemmas D.1, D.2, D.3, D.4 and D.5.

We have the reformulated gradient from Lemma C.4 as follows.

$$\frac{\mathrm{dLoss}(x)}{\mathrm{d}x} = \underbrace{\widetilde{\mathsf{A}}^\top}_{d^4 \times n^2} \mathrm{vec}(\underbrace{\gamma(x)}_{n \times n})$$

Therefore, $n^{1+o(1)}$ is the total running time.

We show that

$$\|\frac{\mathrm{dLoss}(X)}{\mathrm{d}x} - \widetilde{g}\|_\infty = \| \underbrace{\widetilde{\mathsf{A}}^\top}_{d^4 \times n^2} \mathrm{vec}(\underbrace{\gamma(x)}_{n \times n}) - \underbrace{\widetilde{\mathsf{A}}^\top}_{d^4 \times n^2} \mathrm{vec}(\underbrace{\widetilde{\gamma}(x)}_{n \times n})\|_\infty$$

$$= \| \underbrace{\widetilde{\mathsf{A}}^\top}_{d^4 \times n^2} (\mathrm{vec}(\underbrace{\gamma(x)}_{n \times n}) - \mathrm{vec}(\underbrace{\widetilde{\gamma}(x)}_{n \times n}))\|_\infty$$

$$= \| \underbrace{\widetilde{\mathsf{A}}^\top}_{d^4 \times n^2} \|_\infty \| \mathrm{vec}(\underbrace{\gamma(x)}_{n \times n}) - \mathrm{vec}(\underbrace{\widetilde{\gamma}(x)}_{n \times n})\|_\infty$$

$$= \| \underbrace{\widetilde{\mathsf{A}}^\top}_{d^4 \times n^2} \|_\infty \|\gamma(x) - \widetilde{\gamma}(x)\|_\infty$$

$$\leq \epsilon/\mathrm{poly}(n).$$

where the first equality is based on Lemma C.4, the second equality is due to the distributive law, the third equality derives from the definition of $\ell_\infty$ norm, the fourth equality is due to the def. of vectorization, and the fifth inequality derives from the Lemmas in Lemma D.4 and Lemma D.5.

We choose $\epsilon = \frac{1}{\mathrm{poly}(n)}$.

Thus, we have finished our proof. $\qquad\square$

**Remark D.7.** *The assumption in Theorem 5.7 is practical. In practice, especially in recent long context tasks, the $n$ is large, e.g., $n = 2 \times 10^6$ for Google's Gemini 1.5 Pro (Gemini, 2024), while the model training uses a half-precision floating-point format, e.g., the bit number is $16$. Furthermore, our assumption is "tight", where if we slightly weaken the assumption, there is no algorithm that can solve the RoPE attention gradient computation in truly sub-quadratic complexity (Theorem 6.1).*

Our Theorem 5.7 accurately approximates ($\epsilon = 1/\mathrm{poly}(n)$) the RoPE attention gradient computation in almost linear time $n^{1+o(1)}$ under practical assumptions (see the above Remark D.7). Thus, our methods solve the last puzzle of RoPE attention acceleration. Combined with previous work on RoPE attention inference (see Lemma 5.1), this may make RoPE attention practical as we overcome the theoretical quadratic time complexity barrier both in inference and training.

# E  HARDNESS

In this section, we provide the lower bound results to compute the gradient of RoPE attention.

**Theorem E.1** (Lower bound). *Assuming* SETH*, for any $q > 0$, for the* ARAttLGC$(n, d = O(\log n), B = \omega(\sqrt{\log n})$*, there does not exist an algorithm which can be executed in time $O(n^{2-q})$ based on Def. 3.3.*

*Proof.* We pick all of the $W_{-(n-1)}, \dots, W_{n-1} \in \mathbb{R}^{d \times d}$ as an identity matrix $I_d$. Therefore, the gradient computation of RoPE attention can be treated as the gradient computation of classic attention. Thus, our lower bound result can derive from (Alman & Song, 2024c). $\qquad\square$

## LLM USAGE DISCLOSURE

LLMs were used only to polish language, such as grammar and wording. These models did not contribute to idea creation or writing, and the authors take full responsibility for this paper's content.

