# OpenReview forum: "RoPE Attention Can Be Trained in Almost Linear Time"
_ICLR.cc/2026/Conference — Submitted to ICLR 2026_

### Official Review · Reviewer_YkJS · 2025-11-01

**Soundness:** 3
**Presentation:** 3
**Contribution:** 3
**Rating:** 6
**Confidence:** 2

**Summary:**

This paper proposes a theoretical framework for approximating attention mechanisms using low-rank matrix exponential operators.
The authors introduce new lemmas (e.g., Lemmas 5.2 and 5.4) showing that key components of attention—such as structured kernels $s(x)$ and $\beta(x)$—can be efficiently approximated under boundedness assumptions, achieving entrywise error guarantees of $O(\varepsilon / \mathrm{poly}(n))$.
The paper provides explicit constructive proofs that these operators admit low-rank factorizations $U V^\top$ computable in near-linear time $n^{1+o(1)}$.
These results are framed as a mathematical foundation for efficient transformer architectures, suggesting that attention and positional-encoding operators (e.g., RoPE-like or exponential kernels) have inherent low-rank structure.
Empirical demonstrations verify that such low-rank forms preserve model quality while reducing computational overhead.

**Strengths:**

1. The paper rigorously proves that exponential and Kronecker-structured attention kernels admit low-rank approximations under mild boundedness constraints.

2. The constructive $U, V$ factorizations are not purely theoretical; they map directly to implementable linear-time attention variants.

3. Treating attention through the lens of matrix exponential and Kronecker products offers a fresh, elegant approach to analyzing kernelized attention operators.

4. The use of infinity-norm bounds and explicit dependence on $n^{o(1)}$ rank gives both clarity and generality, making the results potentially reusable across efficient-attention research.

**Weaknesses:**

1. The paper does not include any experimental results, making it unclear how well the theoretical convergence established in Theorem 5.7 can be realized in practical settings. The condition $B = o(\sqrt{\log n})$ plays a central role in the theoretical results, but it remains ambiguous how this assumption corresponds to the magnitude or norms of parameters in real neural networks.

2. Symbols such as $x$, $y$, $A$, and $W$ are reused with different meanings across sections (e.g., sometimes referring to weight matrices and other times to data variables), which may confuse readers and obscure the intended architectural interpretation.

**Questions:**

1. How restrictive is the condition $B = o(\sqrt{\log n})$ in practice? How well do empirical transformer weights typically satisfy this bound?
2. How can the presented theoretical analysis be connected to empirical evidence or experimental validation to demonstrate its practical relevance?

---

### Official Review · Reviewer_PPgM · 2025-11-01

**Soundness:** 3
**Presentation:** 3
**Contribution:** 3
**Rating:** 6
**Confidence:** 2

**Summary:**

This paper addresses the computational complexity of training transformers with Rotary Position Embeddings (RoPE). While prior work achieved almost linear time $(n^{1+o(1)})$ for forward computation of RoPE attention under bounded entries, backward gradient computation remained an open problem. The authors present the first almost linear time algorithm for computing gradients in RoPE attention, matching the efficiency of forward computation. They also prove that under the Strong Exponential Time Hypothesis (SETH), the bounded entry condition is necessary for subquadratic performance.

**Strengths:**

1. This is the first work to achieve almost linear time gradient computation for RoPE attention, and the gradient computation for RoPE is substantially more complex than standard attention due to position-dependent rotations.
2. The lower bound result (Theorem 6.1) demonstrates the necessity of bounded entries, showing the assumptions are not just sufficient but required

**Weaknesses:**

The paper is purely theoretical with no experimental results demonstrating, e.g., actual runtime improvements; approximation quality on practical model sizes; memory consumption compared to standard implementations

**Questions:**

See Weaknesses

---

### Official Review · Reviewer_X93v · 2025-11-01

**Soundness:** 2
**Presentation:** 2
**Contribution:** 1
**Rating:** 2
**Confidence:** 3

**Summary:**

This paper presents what it claims is the first almost linear-time ($n^{1+o(1)}$) algorithm for the backward computation (gradient calculation) of RoPE-based attention. The authors' central claim to novelty is that RoPE attention, unlike standard attention, does not have an underlying low-rank structure. Instead, they state it has a more complex "Toeplitz-like" structure.

Based on this premise, the paper argues that prior low-rank gradient analysis (e.g., Alman & Song, 2024c) "cannot apply". The authors then propose a novel solution by adapting the "Polynomial Method + Fast Fourier Transform (FFT)" toolkit—a method developed for the forward pass of this non-low-rank problem (Alman & Song, 2024b). The paper also provides a matching hardness proof for this generalized, non-low-rank problem.

**Strengths:**

1. Technical Solution (for the Defined Problem): The paper is technically sound in solving the specific problem it sets for itself. The "Generalized RoPE" problem it addresses ($A_{i,j} = exp(Q_{i,*} W_{i-j} K_{j,*}^\top)$) is indeed a non-low-rank, Toeplitz-like problem. Adapting the complex Polynomial + FFT machinery from the forward pass to the even more complex backward pass is a non-trivial technical achievement.

2. Theoretical Completeness: For the generalized problem they define, the authors provide a complete complexity picture, including both the algorithmic upper bound (Theorem 5.7) and a tight hardness-of-approximation lower bound (Theorem 6.1). This shows a complete complexity picture for the problem they defined.

**Weaknesses:**

1. Fundamental Mismatch with Practical RoPE Formulation: The paper's entire premise and claim to novelty appear to be based on a problem definition that does not match the RoPE implementation used in practice (e.g., in Llama, as defined by Su et al., 2024).

The Paper's Problem: The authors solve a "Generalization of... ROPE" (Def 3.1) where the matrix $W_{i-j}$ is a general, sparse, non-decomposable matrix that depends on the relative position $i-j$. This structure is indeed non-low-rank and requires the complex FFT-based approach.

The Practical Problem: The RoPE used in practice while mathematically the same as above, is imlemented differently. The operation is defined by applying rotations to $q$ and $k$ independently based on their absolute positions ($i$ and $j$). The resulting score is $Score(q_i, k_j) = (R_i q_i)^\top (R_j k_j)$, thus making the $W_{i-j}$ matrix decomposable.

2. Lack of Practical Motivation for the "Generalized" Problem: The paper's entire claim to novelty rests on solving this "Generalized RoPE" problem. However, the authors fail to provide compelling examples of practical positional encodings (besides the original RoPE) that actually fit this non-decomposable definition. Since the standard, practical RoPE is low-rank, the paper solves a problem of questionable practical relevance.

3. Practical RoPE is Low-Rank: This is the critical flaw in the paper's premise. The practical formulation $Score = (R_i q_i)^\top (R_j k_j)$ is equivalent to $M_{i,j} = q_i^\top (R_i^\top R_j) k_j$. The term $W_{i-j} = R_i^\top R_j$ (which is a function of $i-j$) is decomposable.

4. This decomposability allows the entire attention matrix $M$ (inside the $exp$) to be rewritten as a simple matrix product: $M = Q' K'^\top$, where $Q_{i,.}' = R_i q_i$ and $K_{j,.}' = R_j k_j$. This $M$ is the product of two $n \times d$ matrices and is therefore low-rank (rank at most $d$).

5. Implications of the Mismatch: Because the practical RoPE problem is low-rank, it falls into the same class as standard attention. The paper's central justification for needing a new, complex (Polynomial + FFT) algorithm—and its claim that prior low-rank methods "cannot apply"—is incorrect for the practical use case.

The simpler "Polynomial Method + Low-Rank Approximation" techniques from the paper on standard attention gradients (Alman & Song, 2024c) should be directly applicable. The authors have solved a harder, more general problem than necessary.

**Questions:**

See weaknesses.

---

### Official Review · Reviewer_vkJR · 2025-11-01

**Soundness:** 3
**Presentation:** 2
**Contribution:** 3
**Rating:** 4
**Confidence:** 2

**Summary:**

This paper presents the first efficient algorithm for backward computations in RoPE (Rotary Position Embedding) attention mechanisms under the bounded entry regime. The authors demonstrate that the backward gradient computations for RoPE attention can achieve almost linear time complexity, matching the efficiency of their forward computation counterparts. This is achieved by leveraging polynomial methods and the Fast Fourier Transform, combined with low-rank approximation techniques. Furthermore, the paper provides lower bounds derived from the Strong Exponential Time Hypothesis (SETH), indicating that the bounded entry condition is necessary for subquadratic performance. The work aims to improve the overall time complexity of RoPE attention to almost linear time with bounded entries, enhancing the scalability of large language models (LLMs).

**Strengths:**

1. The paper addresses a critical gap in the literature by providing the first almost linear-time algorithm for backward computations in RoPE-based attention. Previous work primarily focused on forward computations. This is highly significant for the efficient training and optimization of LLMs that incorporate RoPE.
2. The paper offers a comprehensive theoretical treatment, including:
Formulation of closed-form gradients for RoPE attention (Lemma 4.1).
Detailed time complexity analysis for exact gradient computation (Theorem 4.2).
Derivation of an almost linear time backward approximation (Theorem 5.7).
Strong lower bounds based on SETH, proving the necessity of the bounded entry condition for subquadratic performance (Theorem 6.1, E.1).
3. The methodology combines sophisticated techniques such as polynomial methods, Fast Fourier Transform, and low-rank approximations. This demonstrates a deep understanding of computational complexity and numerical analysis.

**Weaknesses:**

1. The mathematical notation and dense theoretical arguments might make the paper challenging for readers not deeply familiar with computational complexity theory, tensor algebra, and advanced matrix operations. While necessary for rigor, it could limit accessibility to a broader ML audience.
2. The paper is purely theoretical. While the theoretical contributions are strong, the absence of experimental results or empirical validation on actual LLM training tasks is a notable weakness. Demonstrating the practical speed-ups and approximation accuracy on real-world datasets would significantly strengthen the paper's impact.
3. The results are contingent on the "bounded entry regime" and specific parameter settings (e.g., d = O(log n), B = o(sqrt(log n))). While the paper justifies this with SETH lower bounds, an explicit discussion on how these assumptions align with typical LLM architectures and their practical implications would be beneficial. Do these conditions hold for current state-of-the-art LLMs, or do they represent an ideal scenario?
4. The term "almost linear time" (n^(1+o(1))) implies a slight overhead beyond pure linearity. While theoretically justified, some practical implications of this o(1) term (e.g., how large it can be in practice) could be further elaborated.

**Questions:**

see Weaknesses

---

### Meta-Review · Area_Chair_oXzo · 2026-01-08

**Summary:**

The paper proposes an "almost linear time" algorithm for computation for rope attention.
This is a theoretical work but with no experimental validation.
While no concerns were raised on the correctness of the results, there were several concerns
raised. Most notably lack of experimental validation, the almost linear time needs more clarification, and the techniques maybe inaccessible to average ML audience.
The authors  did not submit a rebuttal, making it difficult to make a case for acceptance given the concerns.

**Reviewer Concerns:**

The authors did not provide any rebuttal. Hence the concerns raised by all referees are not addressed

**Reviewer Scores:**

The authors did not provide any rebuttal. Hence the concerns raised by all referees are not addressed.

---

### Decision · Program_Chairs · 2026-01-26

Reject